# Surrogate Objectives for Batch Policy Optimization in One-step Decision Making

**Minmin Chen**[*]   **Ramki Gummadi**[*]   **Chris Harris**[*]   **Dale Schuurmans**[*][†]

[*]Google                              [†]University of Alberta

## Abstract

We investigate batch policy optimization for cost-sensitive classification and contextual bandits—two related tasks that obviate exploration but require generalizing from observed rewards to action selections in unseen contexts. When rewards are fully observed, we show that the expected reward objective exhibits suboptimal plateaus and exponentially many local optima in the worst case. To overcome the poor landscape, we develop a convex surrogate that is calibrated with respect to entropy regularized expected reward. We then consider the partially observed case, where rewards are recorded for only a subset of actions. Here we generalize the surrogate to partially observed data, and uncover novel objectives for batch contextual bandit training. We find that surrogate objectives remain provably sound in this setting and empirically demonstrate state-of-the-art performance.

## 1  Introduction

Cost-sensitive classification [1] and batch contextual bandits [34–36] are two problems that share the goal of inferring, given a batch of training data, a *policy* that chooses high reward actions in potentially unseen contexts. The problems differ in the assumed completeness of the data: in cost-sensitive classification, rewards are given (or inferable [8]) for every action, whereas in batch contextual bandits, rewards are only observed for a small subset of actions (typically one). The batch contextual bandit problem is more prevalent in practice, since massive data logs routinely record contexts encountered, actions taken in response, and the outcomes that resulted [18]. Rarely, if ever, are counterfactual outcomes recorded for actions that might have been taken instead [3]. Nevertheless, we find it helpful to reconsider cost-sensitive classification, since a core learning challenge is orthogonal to reward incompleteness: both tasks create difficult optimization landscapes.

There is an extensive literature on cost-sensitive classification. Problems with two actions have been particularly well studied [5, 8], and subsequent work has sought to reduce multiple-action learning to learning binary decisions [1, 19]. A reduction strategy has also been used to convert simply trained stochastic policies to cost-sensitive variants via post-processing [24]. Unfortunately, such reductions do not compose well with current policy learning methods, which are gradient based and best formulated as optimizing a single policy model over a well formed optimization objective.

In this paper we investigate cost-sensitive classification with stochastic policy representations, to ensure the developments are compatible with current deep learning methods. Our first result is negative: for natural policy representations, the expected reward objective generates a poor optimization landscape that exhibits plateaus and potentially an exponential number of local maxima. In response, we develop surrogate objectives for training [28]. Supervised learning research has observed that solution quality can be ensured by using surrogates that satisfy "calibration" with respect to a difficult to optimize target loss [32, 37, 42]. This idea has also recently been applied to cost-sensitive classification [26]. We extend this approach to stochastic policies and deep models by considering expected reward augmented with entropy regularization. This allows a convex surrogate to be developed that improves trainability while approximating expected cost to controllable accuracy.

We then consider batch contextual bandits, where rewards are observed only for a subset (typically one) of the available actions in each training context. Current work has focused on direct maximization of expected reward, using importance correction to provide unbiased (or nearly unbiased) estimates of target gradients [14, 16, 41, 43]. Unfortunately, as we illustrate, such an objective creates an extremely difficult optimization landscape, even if variance can be reduced to *zero* [6, 9]. Alternatively, we extend the calibrated surrogate to partially observed rewards, through the introduction of imputed estimates. We prove soundness of the approach and demonstrate empirical performance benefits.

## 2 Cost-sensitive Classification

We first consider cost-sensitive classification. For simplicity assume a finite set of actions $A = \{1, ..., K\}$, and thus we are given training data $\mathcal{D} = \{(x_i, \boldsymbol{r}_i)\}_{i=1}^{T}$, where $\boldsymbol{r}_i \in \mathbb{R}^K$ is a vector that specifies the reward for each action in context $x_i$. The goal is to infer a mapping $h : X \to A$ that specifies a high reward action $a \in A$ for a given context $x \in X$. **Notation:** We let $\Delta^K$ denote the $K$ dimensional simplex, $\mathbf{1}$ the vector of all 1s, and $\mathbf{1}_a$ the vector of 0s except for 1 in coordinate $a$.

Much of the literature on cost-sensitive classification has focused on deterministic classifiers $h$, but we consider stochastic policies $\boldsymbol{\pi} : X \to \Delta^K$. Any deterministic classifier $h$ can be equivalently expressed by $\boldsymbol{\pi}(x) = \mathbf{1}_{h(x)}$. We seek a policy that maximizes expected reward, or equivalently minimizes expected cost. If we assume the data source is i.i.d. with a joint distribution $p(x, \boldsymbol{r})$, the *true risk* of a policy $\boldsymbol{\pi}$ and its *empirical risk* on data set $\mathcal{D}$ can be defined respectively by

$$\mathcal{R}(\boldsymbol{\pi}) \;\; = \;\; -\mathbb{E}[\boldsymbol{\pi}(x) \cdot \boldsymbol{r}] \quad \text{and} \quad \hat{\mathcal{R}}(\boldsymbol{\pi}, \mathcal{D}) \;\; = \;\; -\tfrac{1}{T} \sum_{(x_i, \boldsymbol{r}_i) \in \mathcal{D}} \boldsymbol{\pi}(x_i) \cdot \boldsymbol{r}_i. \tag{1}$$

Since expected cost is the target, one might presume that directly minimizing empirical risk would be a reasonable approach; unfortunately, this proves problematic [13]. In practice, it is nearly universal to train an unconstrained model $\boldsymbol{q} : X \to \mathbb{R}^K$ that is converted to a policy via a "softmax" transfer; that is, policies are normally represented with the composition $\boldsymbol{\pi}(x) = \boldsymbol{f}(\boldsymbol{q}(x))$ where the model output $\boldsymbol{q}(x)$ is converted to a probability vector via

$$\boldsymbol{f}(\boldsymbol{q}) = e^{\boldsymbol{q} - F(\boldsymbol{q})} \quad \text{with} \quad F(\boldsymbol{q}) = \log(\mathbf{1} \cdot e^{\boldsymbol{q}}). \tag{2}$$

The true and empirical risk can then be re-expressed in terms of $\boldsymbol{q}$ by

$$\mathcal{R}(\boldsymbol{f} \circ \boldsymbol{q}) \;\; = \;\; -\mathbb{E}[\boldsymbol{f}(\boldsymbol{q}(x)) \cdot \boldsymbol{r}] \quad \text{and} \quad \hat{\mathcal{R}}(\boldsymbol{f} \circ \boldsymbol{q}, \mathcal{D}) \;\; = \;\; -\tfrac{1}{T} \sum_{(x_i, \boldsymbol{r}_i) \in \mathcal{D}} \boldsymbol{f}(\boldsymbol{q}(x_i)) \cdot \boldsymbol{r}_i. \tag{3}$$

Unfortunately, the dot product $\boldsymbol{r} \cdot \boldsymbol{f}(\boldsymbol{q}(x))$ creates significant difficulty, as this interacts poorly with the softmax transfer $\boldsymbol{f}$. A well known consequence is that the expected cost plateaus whenever the corresponding policy probabilities are nearly deterministic. A potentially greater challenge, however, is that the softmax transfer can also induce exponentially many local optima.

**Theorem 1** *Even for a single context $x$, a deterministic reward vector $\boldsymbol{r}$, and a linear model $\boldsymbol{q}(x) = W\phi(x)$, the function $\boldsymbol{r} \cdot \boldsymbol{f}(\boldsymbol{q}(x))$ can have a number of local maxima in $W$ that is exponential in the number of actions $K$ and the number of features in $\phi$. (**All proofs given in the appendix.**[1])*

It is therefore unsurprising that empirical risk minimization with stochastic policies is not considered viable in the cost-sensitive classification literature. Nevertheless, it remains the dominant approach for batch contextual bandits. We seek to bridge the apparent disconnect between these two settings.

### 2.1 Calibrated Strongly Convex Surrogate

A key idea in cost-sensitive classification has been the development of convex surrogate objectives that exhibit "calibration" with respect to the target risk [2, 32]. We require additional definitions. Let $\mathcal{Q}$ denote the set of measurable functions $X \to \mathbb{R}^K$, and define the minimum risk by $\mathcal{R}^* = \inf_{\boldsymbol{q} \in \mathcal{Q}} \mathcal{R}(\boldsymbol{f} \circ \boldsymbol{q})$. Note that the minimum is generally achieved at a deterministic policy, which cannot be represented by $\boldsymbol{q} \in \mathcal{Q}$; however, the infimum can be arbitrarily well approximated within $\mathcal{Q}$. It will be convenient to expand the risk definition through a notion of pointwise risk: define the *local risk* as $\mathcal{R}(\boldsymbol{\pi}, \boldsymbol{r}, x) = -\boldsymbol{r} \cdot \boldsymbol{\pi}(x)$, which is related to the true risk via $\mathcal{R}(\boldsymbol{\pi}) = \mathbb{E}[\mathcal{R}(\boldsymbol{\pi}, \boldsymbol{r}, x)]$, with the expectation taken over pairs $(x, \boldsymbol{r}) \sim p(x, \boldsymbol{r})$. For each $(x, \boldsymbol{r})$ define the minimal risk by

$$\mathcal{R}^*(\boldsymbol{r}, x) \;\; = \;\; \inf_{\boldsymbol{\pi} \in \mathcal{P}} \mathcal{R}(\boldsymbol{\pi}, \boldsymbol{r}, x) \;\; = \;\; \inf_{\boldsymbol{q} \in \mathcal{Q}} \mathcal{R}(\boldsymbol{f} \circ \boldsymbol{q}, \boldsymbol{r}, x). \tag{4}$$

Consider a surrogate loss function $L : (\mathcal{Q}, \mathbb{R}^K, X) \to \mathbb{R}$ and let $L^*(\boldsymbol{r}, x) = \inf_{\boldsymbol{q} \in \mathcal{Q}} L(\boldsymbol{q}, \boldsymbol{r}, x)$. We say that a surrogate $L$ is *calibrated* with respect to the target risk $\mathcal{R}$ if there exists a calibration function $\delta(\epsilon, x) \geq 0$ such that for all $\epsilon > 0$, all $x \in X$, all $\boldsymbol{r} \in \mathbb{R}^K$ and all $\boldsymbol{q} \in \mathcal{Q}$:

$$L(\boldsymbol{q}, \boldsymbol{r}, x) - L^*(\boldsymbol{r}, x) \; < \; \delta(\epsilon, x) \quad \text{implies} \quad \mathcal{R}(\boldsymbol{f} \circ \boldsymbol{q}, \boldsymbol{r}, x) \; < \; \mathcal{R}^*(\boldsymbol{r}, x) + \epsilon. \tag{5}$$

Although calibrated convex surrogates have been developed for cost-sensitive classification [26], these do not consider stochastic policies. Rather than extending these constructions to stochastic policies, which is not straightforward, we develop a new surrogate for the stochastic case. Consider an entropy regularized version of the target risk [25] which we call the *smoothed risk*:

$$\mathcal{S}_\tau(\boldsymbol{\pi}, \boldsymbol{r}, x) \;\; = \;\; -\boldsymbol{r} \cdot \boldsymbol{\pi}(x) + \tau \boldsymbol{\pi}(x) \cdot \log \boldsymbol{\pi}(x) \quad \text{and} \quad \mathcal{S}_\tau(\boldsymbol{\pi}) \;\; = \;\; \mathbb{E}[\mathcal{S}_\tau(\boldsymbol{\pi}, \boldsymbol{r}, x)]. \tag{6}$$

The smoothed risk approximates the true risk, with a discrepancy that can be made arbitrarily small.

**Proposition 2** *Let $\tilde{\boldsymbol{\pi}}_\tau = \arg\min_{\boldsymbol{\pi} \in \mathcal{P}} \mathcal{S}_\tau(\boldsymbol{\pi})$. Then $\tilde{\boldsymbol{\pi}}_\tau(x) = \exp(\mathbb{E}[\boldsymbol{r}|x] - F(\mathbb{E}[\boldsymbol{r}|x])/\tau)$ and $\mathcal{R}(\tilde{\boldsymbol{\pi}}_\tau) < \mathcal{R}^* + \tau \log K$. Hence for any $\epsilon > 0$ setting $\tau < \epsilon/\log K$ ensures $\mathcal{R}(\tilde{\boldsymbol{\pi}}_\tau) < \mathcal{R}^* + \epsilon$.*

Note that the smoothed risk is not convex in $\boldsymbol{q}$ due to the softmax transfer $\boldsymbol{\pi}(x) = \boldsymbol{f}(\boldsymbol{q}(x))$. Nevertheless, it is possible to develop a convex surrogate that is calibrated for the smoothed risk as follows. First we need a few properties of Bregman divergences in general and the KL divergence in particular. The Bregman divergence $D_F$, specified by the convex differentiable potential $F$, satisfies:

$$D_F(\boldsymbol{q}\|\boldsymbol{r}) \;\; = \;\; F(\boldsymbol{q}) - F(\boldsymbol{r}) - \boldsymbol{f}(\boldsymbol{r}) \cdot (\boldsymbol{q} - \boldsymbol{r}) \;\; = \;\; F(\boldsymbol{q}) - \boldsymbol{q} \cdot \boldsymbol{p} + F^*(\boldsymbol{p}) \;\; = \;\; D_{F^*}(\boldsymbol{p}\|\boldsymbol{\pi}), \tag{7}$$

where $\boldsymbol{f} = \nabla F$, $F^*(\boldsymbol{p})$ is the convex conjugate of $F$, $\boldsymbol{p} = \boldsymbol{f}(\boldsymbol{r})$ and $\boldsymbol{\pi} = \boldsymbol{f}(\boldsymbol{q})$ [27]. Clearly, $D_F$ is convex in its first argument $\boldsymbol{q}$, but not necessarily in the second. For the KL divergence in particular we have $F(\boldsymbol{q}) = \log \boldsymbol{1} \cdot e^{\boldsymbol{q}}$, $\boldsymbol{f}(\boldsymbol{q}) = e^{\boldsymbol{q} - F(\boldsymbol{q})}$, $F^*(\boldsymbol{p}) = \boldsymbol{p} \cdot \log \boldsymbol{p}$, hence

$$D_{KL}(\boldsymbol{\pi}\|\boldsymbol{p}) = \boldsymbol{\pi} \cdot (\log \boldsymbol{\pi} - \log \boldsymbol{p}) = D_{F^*}(\boldsymbol{\pi}\|\boldsymbol{p}) = D_F(\boldsymbol{r}\|\boldsymbol{q}). \tag{8}$$

This means that the local smoothed risk (6) can be shown to be equivalent to

$$\mathcal{S}_\tau(\boldsymbol{\pi}, \boldsymbol{r}, x) \;\; = \;\; -\tau\left(\tfrac{\boldsymbol{r}}{\tau} \cdot \boldsymbol{\pi}(x) - \boldsymbol{\pi}(x) \cdot \log \boldsymbol{\pi}(x)\right) = -\tau F(\tfrac{\boldsymbol{r}}{\tau}) + \tau D_F\left(\tfrac{\boldsymbol{r}}{\tau}\|\boldsymbol{q}(x)\right). \tag{9}$$

Later, in Section 3, we will find it helpful to consider a shift $v$ of the expected cost; i.e. $\mathcal{R}(\boldsymbol{\pi}, \boldsymbol{r} - v, x) = v - \boldsymbol{r} \cdot \boldsymbol{\pi}$, noting this does not affect the location of the minimizer in $\boldsymbol{q}$. The above characterization then allows us to formulate a convex calibrated surrogate by reversing the divergence.

**Theorem 3** *For an arbitrary baseline $v$ and $\tau > 0$, let*

$$L(\boldsymbol{q}, \boldsymbol{r}, x) = \tau D_F\left(\boldsymbol{q}(x) + \tfrac{v}{\tau} \,\big\|\, \tfrac{\boldsymbol{r}}{\tau}\right) + \tfrac{\tau}{4}\left\|\boldsymbol{q}(x) - \tfrac{\boldsymbol{r}-v}{\tau}\right\|^2. \tag{10}$$

*Then, for any fixed $v$, $L$ is strongly convex in $\boldsymbol{q}$ and calibrated with respect to the smoothed (shifted) risk $\mathcal{S}_\tau(\boldsymbol{f} \circ \boldsymbol{q}, \boldsymbol{r} - v, x) = \mathcal{S}_\tau(\boldsymbol{f} \circ \boldsymbol{q}, \boldsymbol{r}, x) - v$ with calibration function $\delta(\epsilon, x) = \epsilon \; \forall x$.*

Therefore, any desired level of accuracy in minimizing empirical smoothed risk can be achieved by approximately minimizing the surrogate loss $L$ to appropriate accuracy.

## 2.2 Experimental Evaluation

To first assess the overall approach, we evaluate how well optimizing the surrogate (10) minimizes true risk, using a separate test set for evaluation. As baselines, we compare to directly minimizing empirical risk $\hat{\mathcal{R}}(\boldsymbol{\pi})$ (1), and the standard supervised objectives, log-likelihood, $-\mathbb{E}_{\boldsymbol{p}}[\log \boldsymbol{\pi}]$, and squared error, $\|\boldsymbol{q}(x) - \tfrac{\boldsymbol{r}-v}{\tau}\|^2$. Empirically, we found it beneficial to relax (10) to a tunable combination between the components and empirical risk. We refer to such a tuned loss as "Composite" in all experimental results. Since the surrogate objective is a combination of the "reversed KL" objective $D_{F^*}(\boldsymbol{p}\|\boldsymbol{\pi})$ (7) and the squared error, we also evaluate $D_{F^*}(\boldsymbol{p}\|\boldsymbol{\pi})$ alone to isolate its effect.

**MNIST** We first consider MNIST data, training a fully connected model with one hidden layer of 512 ReLU units. The original training data was partitioned into the first 55K examples for training and the last 5K examples for validation. We use the validation data to select hyperparameters, including learning rate, mini-batch size, and combination weights (details in appendix). The policy was trained by minimizing each objective using SGD with momentum fixed at 0.9 [33] for 100 epochs.

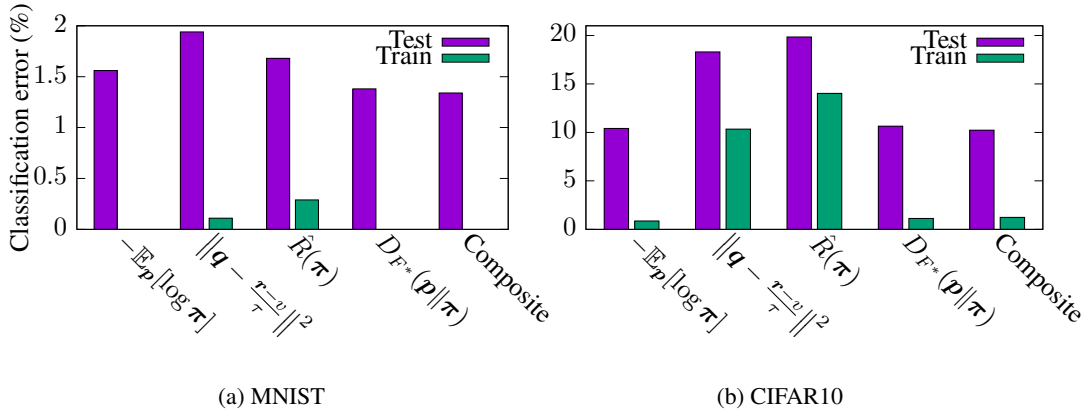

|  | (a) MNIST | (b) CIFAR10 |
|---|---|---|

Figure 1: Training with full reward feedback across all actions (see appendix for additional results).

**CIFAR-10** Next we considered the CIFAR-10 data set [15] and trained a Resnet-20 architecture [12], using the standard 50K training, 10K validation split. We set any unspecified model hyperparameters to the defaults for resnet in the open source tensor2tensor library [39] and tuned learning rate and the composite loss combination weights on validation data. All objectives were trained using the Momentum optimizer with cosine decay learning rate for 250 epochs (details in appendix).

The results in Figure 1 confirm that directly minimizing $\hat{\mathcal{R}}(\boldsymbol{\pi})$ (1), is not always competitive: it yields the highest training error on both MNIST and CIFAR10, as well as poor test error. For MNIST, it is striking that generalization can still be improved with respect to the standard log-likelihood baseline. For CIFAR-10, as shown in Figure 1b, minimizing the reverse KL $D_{F^*}(\boldsymbol{p}\|\boldsymbol{\pi})$ achieved 10.7% test error, which significantly improves directly optimizing empirical risk $\hat{\mathcal{R}}(\boldsymbol{\pi})$, which obtained 19.9% test error. The reverse KL was competitive even against the baseline log-likelihood, which achieved 10.4% test error. The results for squared error are worse than log-likelihood, while the $D_{F^*}(\boldsymbol{p}\|\boldsymbol{\pi})$ objective performs better in both data sets. This suggests that the generalization improvements are coming from better minimization of $D_{F^*}(\boldsymbol{p}\|\boldsymbol{\pi})$, while the squared error term is helping improve the optimization landscape. To further investigate whether $\hat{\mathcal{R}}(\boldsymbol{\pi})$ suffers from a difficult optimization landscape, we ran a much longer training experiment (see appendix), finding that every method except squared loss is eventually able to achieve about 6.5% test error, but at significant cost.

## 3 Batch Contextual Bandits

We now extend these developments to the contextual bandit case. To focus on the most challenging and practical scenario, we assume a single action has been observed in each context. Therefore, the training data consists of tuples $\mathcal{D} = \{(x_i, a_i, r_i, \beta_i)\}$, where $x_i \in X$ is a context, $a_i \in \{1, ..., K\}$ is an action, $r_i \in \mathbb{R}$ is a reward, and $\beta_i$ is the proposal probability of $a_i$. For simplicity, we assume a stationary behaviour (logging) policy $\boldsymbol{\beta} : X \rightarrow \Delta^K$ was used to select the actions, hence $\beta_i = \beta(a_i|x_i)$. Although $\beta$ might not be known [20], estimating it from $\mathcal{D}$ has proved effective [41, 43, 4]. We continue to assume contexts and rewards are generated i.i.d. from a joint distribution $p(x, \boldsymbol{r})$, but the distribution of rewards $\boldsymbol{r}(x) \sim p(\boldsymbol{r}|x)$ and actions $a \sim \beta(a|x)$ are conditionally independent given the context $x$ [43]. Other more elaborate models for missing data are possible, but require committing to stronger assumptions about the data generation and behavior process [3, 21].

As before, the goal is to infer a policy $\boldsymbol{\pi} : X \rightarrow \Delta^K$ that maximizes expected reward. Here we define the *true risk* of a policy $\boldsymbol{\pi}$ as in (1), but the empirical risk, also defined in (1), is no longer directly observable because it requires rewards for all actions. The standard solution is to formulate an unbiased estimate of the full empirical risk (which is itself an unbiased estimate of the true risk), then use this as a policy optimization objective. In fact, the current literature is dominated by such an approach, where an unbiased (or nearly unbiased) estimate of the empirical risk (1) is first formulated via *importance correction* then used as a training objective [14, 16, 17, 29, 34–36]. Unfortunately, importance correction introduces significant variance in gradient estimates, even using standard variance reduction techniques. Also, as identified in Section 2, *even if variance could be completely eliminated*, the underlying optimization landscape presents difficulties.

## 3.1 Reward Estimation

Before focusing on policy optimization, we first need to address the problem of estimating rewards from incomplete data. We here adopt a simple approach of imputing missing values with a model $q : X \to \mathbb{R}^K$. That is, for a context $x$, observed action $a$ and observed reward $r_a$, we estimate the full reward vector $r$ by

$$\hat{r}(x) \;=\; \tau q(x) + \mathbf{1}_a \lambda(x,a)(r_a - \tau q(x)_a), \tag{11}$$

with parameters $\lambda(x,a)$ and $\tau$. This construction allows the local risk of any policy $\pi$ to be estimated by $\mathcal{R}(\pi, \hat{r}, x) = -\pi(x) \cdot \hat{r}(x)$. Although (11) seems simplistic, it is able to express most estimators in the literature by suitable choices of $\tau$ and $\lambda(x,a)$. For example, choosing $\tau = 0$ and $\lambda(x,a) = \beta(a|x)^{-1}$ yields importance weighting $\mathcal{R}(\pi, \hat{r}, x) = \frac{\pi(x)_a}{\beta(a|x)} r_a$; choosing $\tau = 1$ and $\lambda(x,a) = 0$ yields the "direct method" $\mathcal{R}(\pi, \hat{r}, x) = \pi(x) \cdot q(x)$; and choosing $\tau = 1$ and $\lambda(x,a) = \beta(a|x)^{-1}$ yields the "doubly robust" estimate $\mathcal{R}(\pi, \hat{r}, x) = \pi(x) \cdot q(x) + \frac{\pi(x)_a}{\beta(a|x)}(r_a - q(x)_a)$ [6]. The "switch" estimator [40] can be expressed for a given threshold $\theta > 0$ by setting $\tau = 1$ and $\lambda(x,a) = 0$ if $\pi(x)_a/\theta > \beta(a|x)$, otherwise $\tau = 0$ and $\lambda(x,a) = \beta(a|x)^{-1}$. The "switch" estimator generalizes the trimmed importance estimator [3], and is argued in [40] to be superior to the "magic" estimator [38]. The "self-normalized" importance estimator [36] can also be expressed by setting $\tau = 0$ and $\lambda(x,a) = \beta(a|x)^{-1} / \sum_{(x_i,a_i,r_i) \in \mathcal{D}} \pi(x_i)_{a_i} \beta(a_i|x_i)^{-1}$.

For any fixed $q(x)$ and $\tau$ it is easy to show that $\lambda(x,a) = \beta(a|x)^{-1}$ implies $\mathbb{E}_{a \sim \beta(\cdot|x)}[\hat{r}(x)] = r(x)$.

A key question is the provenance of $q$. The standard approach is to recover $q$ by regressing to the observed rewards $q = \arg\min_{q \in \mathcal{H}} \sum_{(x_i,a_i,r_i) \in \mathcal{D}} (r_i - q(x_i)_{a_i})^2$ for a class of models $\mathcal{H}$. Note that this is equivalent to conducting a policy evaluation step for the policy $\beta$. Stochastic contextual bandits are a restricted case of reinforcement learning where every policy has the same action value function, $\mathbb{E}[r(x)]$. Hence, a single policy evaluation, yielding $q$, can in principle be used to evaluate any policy, since evaluating $\pi$ instead of $\beta$ does not change action values but only introduces covariate shift.

## 3.2 Policy Optimization

For policy optimization, one could adopt the least squares estimate $q$ and optimize a separate policy, but if $\pi$ uses the same architecture the optimum is simply $\pi = f \circ q$ (under $\mathcal{S}$). In Section 2, we saw that least squares estimation of $q$ did not perform well, nor do we expect so here. We would like to gain the advantages realized in Section 2, but an actor-critic approach obviates policy optimization.

Instead, to couple the value estimator to policy optimization, we consider a unified approach where the actor and the critic *are the same model*. That is, we use the policy transformation $\pi = f \circ q$ from Section 2, but now explicitly treat the logits as action value estimates. A unified actor-critic model has been considered previously [22]. In the partially observed case, we propose to replace the observed reward vector with the estimate $\hat{r}$ derived from $q$, allowing any loss to be applied. Although such an approach seems naive, we find that maintaining this form of strict mutual consistency between the value estimates and policy, combined with the estimator $\hat{r}$ and surrogate losses, leads to effective empirical performance. Moreover, we will find that this approach is theoretically justified.

## 3.3 Calibrated Surrogate

Given $(x, a, r_a)$, define the optimal imputed local risk and the suboptimality gap respectively by

$$\mathcal{S}^*_\tau(\hat{r}, x) \;=\; \inf_{q \in \mathcal{Q}} \mathcal{S}_\tau(f \circ q, \hat{r}, x) \quad \text{and} \quad \mathcal{G}_\tau(\pi, \hat{r}, x) \;=\; \mathcal{S}_\tau(\pi, \hat{r}, x) - \mathcal{S}^*_\tau(\hat{r}, x). \tag{12}$$

Equality (9) can then be used to show the divergence $D_F\left(\frac{\hat{r}}{\tau} \| q\right)$ characterizes the suboptimality gap:

**Proposition 4** *For any $q$, $\tau > 0$ and observation $(x, a, r_a)$: $\tau D_F\left(\frac{\hat{r}(x)}{\tau} \| q(x)\right) = \mathcal{G}_\tau(f \circ q, \hat{r}, x)$.*

If we consider the imputed form of the surrogate objective $L(q, \hat{r}, x)$ defined in Theorem 3 we then find that the surrogate remains calibrated for the imputed smoothed risk.

**Theorem 5** *For any model $q$, $\tau > 0$, observation $(x, a, r_a)$, and baseline $v$:*

$$L(q, \hat{r}, x) \;\geq\; \tau D_F\left(\frac{\hat{r}(x)}{\tau} \Big\| q(x) + \frac{v}{\tau}\right) \;=\; \mathcal{G}_\tau(f \circ q, \hat{r}, x) \;\geq\; 0. \tag{13}$$

*Moreover, L is calibrated with respect to $\mathcal{S}_\tau(\boldsymbol{f} \circ \boldsymbol{q}, \hat{\boldsymbol{r}} - v, x)$ with calibration function $\delta(x, \epsilon) = \epsilon$.*

This result suggests a simple algorithmic approach for policy optimization: given the data $\mathcal{D} = \{(x_i, a_i, r_i, \beta_i)\}$, minimize the imputed empirical surrogate objective with respect to the model $\boldsymbol{q}$:

$$\min_{\boldsymbol{q} \in \mathcal{Q}} \hat{L}(\boldsymbol{q}, \mathcal{D}) \quad \text{where} \quad \hat{L}(\boldsymbol{q}, \mathcal{D}) = \tfrac{1}{T} \sum_{(x_i, a_i, r_i, \beta_i) \in \mathcal{D}} L(\boldsymbol{q}, \hat{\boldsymbol{r}}, x_i). \quad (14)$$

That is, we combine the estimate $\hat{\boldsymbol{r}}$ from Section 3.1, (11), with the surrogate $L$ from Section 2, (10).

## 3.4 Analysis

The expected smoothed risk quantities we seek to control are defined by:

$$\mathcal{S}_\tau(\boldsymbol{\pi}) = \mathbb{E}[\mathcal{S}_\tau(\boldsymbol{\pi}, \boldsymbol{r}, x)], \quad \mathcal{S}_\tau^* = \inf_{\boldsymbol{q} \in \mathcal{Q}} \mathcal{S}_\tau(\boldsymbol{f} \circ \boldsymbol{q}) \quad \text{and} \quad \mathcal{G}_\tau(\boldsymbol{\pi}) = \mathcal{S}_\tau(\boldsymbol{\pi}) - \mathcal{S}_\tau^*. \quad (15)$$

For the purposes of analysis, we assume training data consists of tuples drawn from $(x, a, r_a) \sim p(x, \boldsymbol{r})\beta(a|x)$, and that the estimate $\hat{\boldsymbol{r}}$ is *unbiased*; i.e., $\mathbb{E}[\hat{\boldsymbol{r}}|x] = \mathbb{E}[\boldsymbol{r}|x]$, using $\lambda(x, a) = \beta(a|x)^{-1}$.

First, observe that, in expectation, the surrogate objective upper bounds the divergence in Theorem 5, which, in turn, by Jensen's inequality, bounds the suboptimality gap in the expected smoothed risk.

**Theorem 6** *For any model $\boldsymbol{q}$, any $\hat{\boldsymbol{r}}$ such that $\mathbb{E}[\hat{\boldsymbol{r}}|x] = \mathbb{E}[\boldsymbol{r}|x]$, and any baseline $v$:*

$$\mathbb{E}[L(\boldsymbol{q}, \hat{\boldsymbol{r}}, x)] \geq \mathbb{E}\left[\tau D_F\left(\tfrac{\hat{\boldsymbol{r}}(x)}{\tau} \Big\| \boldsymbol{q}(x) + \tfrac{v}{\tau}\right)\right] \geq \mathcal{G}_\tau(\boldsymbol{f} \circ \boldsymbol{q}) \geq 0. \quad (16)$$

Therefore, minimizing (14), in expectation, minimizes the true smoothed risk (15).

This result can be made stronger by observing that, under mild assumptions, the empirical divergence $\hat{D}(\boldsymbol{q}, \mathcal{D}) = \tfrac{1}{T} \sum_i D_F\left(\tfrac{\hat{\boldsymbol{r}}(x_i)}{\tau} \big\| \boldsymbol{q}(x_i)\right)$ also concentrates to its expectation, uniformly over $\boldsymbol{q} \in \mathcal{H}$, for a well behaved model class $\mathcal{H}$. **In the appendix**, we specify the conditions on $\mathcal{H}$, $\beta$, and $p(x, \boldsymbol{r})$ that, in addition to $\hat{\boldsymbol{r}}$ being unbiased (i.e. $\mathbb{E}[\hat{\boldsymbol{r}}|x] = \mathbb{E}[\boldsymbol{r}|x]$), ensure finite sample concentration. We refer to a collection $\mathcal{H}$, $\beta$, $p(x, \boldsymbol{r})$ and $\hat{\boldsymbol{r}}$ that satisfies these conditions as "well behaved".

**Lemma 7** *Assume $\mathcal{H}$, $\beta$, $p(x, \boldsymbol{r})$ and $\hat{\boldsymbol{r}}$ are "well behaved". Then for any $\tau, \delta > 0$ there exists a constant $C$ such that with probability at least $1 - \delta$:*

$$\mathbb{E}\left[D_F\left(\tfrac{\hat{\boldsymbol{r}}(x)}{\tau} \Big\| \boldsymbol{q}(x)\right)\right] \leq \hat{D}_F(\boldsymbol{q}, \mathcal{D}) + \tfrac{C}{\sqrt{T}} \quad \forall \boldsymbol{q} \in \mathcal{H}. \quad (17)$$

Combining Theorem 6 with Lemma 7 it can be shown that for finite sample size $T$, with high probability, the empirical surrogate (14) is approximately calibrated with respect to smoothed risk.

**Theorem 8** *Assume $\mathcal{H}$, $\beta$, $p(x, \boldsymbol{r})$ and $\hat{\boldsymbol{r}}$ are "well behaved". Then for any $v$ and $\tau, \delta > 0$, there exists a $C$ such that with probability at least $1 - \delta$: if $\hat{L}(\boldsymbol{q}, \mathcal{D}) < \tfrac{\tau C}{\sqrt{T}}$ for $\boldsymbol{q} \in \mathcal{H}$ then $\mathcal{G}_\tau(\boldsymbol{f} \circ \boldsymbol{q}) \leq \tfrac{2\tau C}{\sqrt{T}}$.*

That is, if $\hat{L}(\boldsymbol{q}, \mathcal{D})$ can be sufficiently minimized within $\mathcal{H}$, the suboptimality gap achieved by $\boldsymbol{q}$ will be near-optimal with high probability, with bound diminishing to zero for large sample size.

## 3.5 Discussion

If we let $\tau = 0$ in the definition of $\hat{\boldsymbol{r}}$, (11), then $\hat{\boldsymbol{r}}$ exhibits no dependence on $\boldsymbol{q}$, making $L(\boldsymbol{q}, \hat{\boldsymbol{r}}, x)$ convex in $\boldsymbol{q}$. However, we have found that empirical results are improved by choosing $\tau > 0$, since this compels the logits $\boldsymbol{q}$ to also model observed rewards. In addition, even though using an unbiased $\hat{\boldsymbol{r}}$ enables the theory above, achieving unbiasedness via importance correction increases variance, degrades the quality of the reward estimate, and yields inferior results. In our experiments, we considered $\tau$ to be a hyperparameter, and also considered different choices for $\lambda$, including $\lambda(x, a) = \beta(a|x)^{-1}$ and $\lambda(x, a) = 1$. We also introduced tunable combination weights between the Bregman divergence and the squared error terms in (14), similar to the relaxation in Section 2.2. In all cases, we chose hyperparameters from validation data only.

Note that the approach developed in this paper differs fundamentally from recent trust-region and proximal methods in reinforcement learning [30, 31], which still directly optimize expected return, possibly with entropy regularization [23]. These methods use proximal constraints/regularization to improve the stability of optimization, but apply a "surrogate" as a local not a global modification of the objective. By contrast, we are changing the entire optimization objective globally, not locally, and train to maximize a target that is different from expected return.

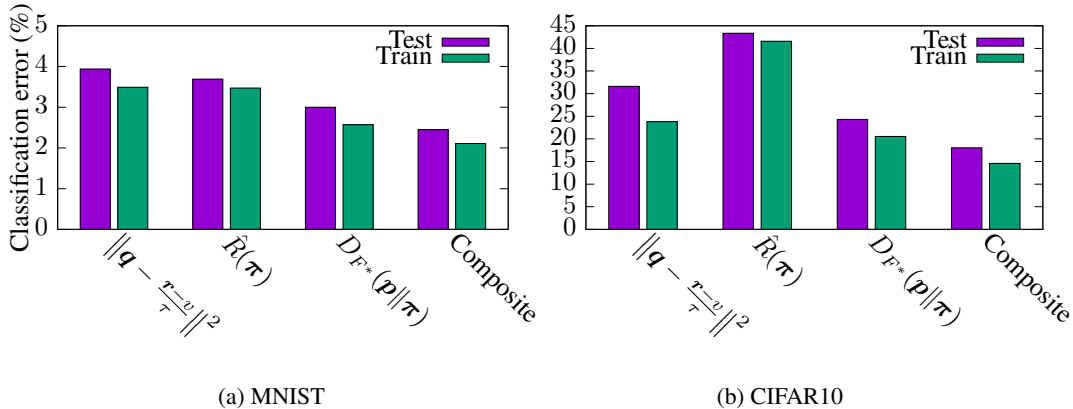

|                   |                   |
|:-----------------:|:-----------------:|
|     (a) MNIST     |    (b) CIFAR10    |

Figure 2: Training with partial (single action) reward feedback (see appendix for additional results).

Unlike previous entropy regularized approaches [11, 22], which generally consider split actor-critic models, we achieve success with a single model that serves as both.

Another subtlety with optimizing importance corrected objectives, such as $\mathcal{R}(\boldsymbol{\pi}, \hat{\boldsymbol{r}}, x) = \frac{\boldsymbol{\pi}(x)_a}{\beta(a|x)} r_a$, is that this does not account for the policy's data coverage [20]; that is, a policy might minimize such an objective by moving mass $\boldsymbol{\pi}(x_i)_{a_i}$ away from the training observations $(x_i, a_i, r_i)$, leading to a phenomenon known as "propensity overfitting" [34–36]. This effect can be countered by adding coverage-dependent confidence intervals to the estimates [34–36], or constraining [10] or regularizing [20] toward the logging policy choices. Although such regularization is helpful, it is orthogonal to the aim of the current investigation, as any objective can be augmented in this way.

### 3.6 Experimental Evaluation

As is standard in the field [34–36], we form a partially observed version of a supervised learning task by sampling actions from a behaviour policy $\boldsymbol{\pi}_0$, assigning a reward of 1 when the action chosen matches the correct label and a reward of 0 when it does not. Reward on all counterfactual actions is therefore missing. For the MNIST and CIFAR-10 experiments, we used the same architecture, optimizer and model configurations used in the fully observed label experiments. For the empirical risk estimator $\hat{\mathcal{R}}(\boldsymbol{\pi})$ we used importance correction $\lambda(x, a) = \beta(a|x)^{-1}$; however $\lambda(x, a) = 1$ proved to be more effective for $D_{F^*}(\boldsymbol{p}\|\boldsymbol{\pi})$, which is equivalent to replacing the counterfactual rewards with the model estimate.

**MNIST** We evaluate the results when data is collected by the uniform behavior policy, $\boldsymbol{\pi}_0(x) = \mathbf{1} \cdot \frac{1}{10}$. The hyperparameters for all objectives were re-optimized on validation data, using the same optimization algorithm as before. Details are given in the appendix.

**CIFAR-10** Here we also evaluate the results when data is collected by the uniform behavior policy, $\boldsymbol{\pi}_0(x) = \mathbf{1} \cdot \frac{1}{10}$. However, in addition, we also evaluate the proposed objectives using data released in a recently published benchmark on CIFAR-10 [14]. (Note that the behavior policy itself was not released in this benchmark; instead different sized training sets of size 50k, 100k, 150k, and 250k were generated using this policy.) We used this alternative data to produce each column in Table 1. For the CIFAR-10 experiments we simply set $\tau = 1$. Additional details are given in the appendix.

Figure 2 shows the results for training on MNIST and CIFAR-10 given data collected by the random behavior policy. In both cases, the composite objective yields improvements over optimizing $\hat{R}(\boldsymbol{\pi})$ directly. To investigate whether this difficulty is due to plateaus, we again conduct significantly longer training in the appendix, finding that the Composite and $D_{F^*}(\boldsymbol{p}\|\boldsymbol{\pi})$ objectives remain advantageous. Table 1 then shows results on CIFAR-10 using the alternative behavior data from [14]. This data appears to be more condusive to optimizing $\hat{R}(\boldsymbol{\pi})$ directly, although even in this scenario the composite objective is still competitive, significantly improving the results reported in [14].

**Criteo** We also test the proposed surrogate objective on the Criteo data set [18], a large-scale test-bed for evaluating batch contextual bandit methods [3, 34]. Here again the behavior policy was not

| Examples | 50k | 100k | 150k | 250k |
|---|---|---|---|---|
| $\hat{\mathcal{R}}(\boldsymbol{\pi})$ | 8.71 | 7.51 | 6.92 | 6.66 |
| $\left\|\boldsymbol{q} - \frac{r-v}{\tau}\right\|^2$ | 21.85 | 18.26 | 14.65 | 12.86 |
| $D_{F^*}(\boldsymbol{p}\|\boldsymbol{\pi})$ | 16.00 | 9.85 | 8.68 | 8.75 |
| **Composite** | 8.34 | 6.92 | 6.57 | 6.36 |

Table 1: CIFAR-10: Test error % for the bandit feedback data sets from [14] with increasing number of training examples.

| Objectives | $\hat{\mathcal{R}}(\pi) \times 10^4$ |
|---|---|
| Random | $43.68 \pm 2.11$ |
| Behavior | $53.55$ |
| DRO $\hat{\mathcal{R}}(\boldsymbol{\pi})$ [7] | $53.07 \pm 2.27$ |
| POEM [34] | $51.89 \pm 1.73$ [2] |
| $\hat{\mathcal{R}}(\boldsymbol{\pi})$ | $51.72 \pm 1.42$ |
| $\left\|\boldsymbol{q} - \frac{r-v}{\tau}\right\|^2$ | $52.00 \pm 1.28$ |
| $D_{F^*}(\boldsymbol{p}\|\boldsymbol{\pi})$ | $52.30 \pm 0.83$ |
| **Composite** | $55.09 \pm 2.86$ |

Table 2: Criteo: Importance sampling estimated reward on test. Error bars are $99\%$ confidence intervals under normal distribution.

released, but only its generated data. Following [18], we use only banners with a single slot (i.e., where only a single item is chosen) in our learning and evaluation. These banners are randomly split into training, validation and test sets, each containing 7 million records, using the script provided by [18]. There are 35 features used to describe the context and candidates actions (2 continuous and the rest categorical). We encode the discrete features using one-hot encoding, and build linear models using different learning losses. For evaluation, we report the importance sampling based estimates of reward (user clicks on banner) $\hat{R}(\boldsymbol{\pi})$ on the test set, as in [18].

We compare the proposed surrogates with several state-of-the-art methods on this data set. Hyperparameters of the different methods were tuned on validation data, and all objectives optimized by SGD with momentum and batch size between 1K and 5K; more details regarding the experiment setup and hyperparameter choices are given in the appendix. All methods use the same input encoding to map inputs $x$ to $\phi(x)$. In particular, we evaluated the following:

**Random**: Choose a candidate banner (i.e. an action) uniformly at random to display.
**Behavior:** Simply report observed reward on the test set acting according to the logging policy $\boldsymbol{\beta}$.
**Squared** $\left\|\boldsymbol{q} - \frac{r-v}{\tau}\right\|^2$: We set $v = 0$ is effective since expected reward is close to 0.
$\hat{\mathbf{R}}(\boldsymbol{\pi})$: Directly optimize $\hat{R}(\boldsymbol{\pi})$ using importance correction; i.e. $\lambda(x,a) = \beta(a|x)^{-1}$, $\tau = 0$ in (11).
**DRO** $\hat{\mathbf{R}}(\boldsymbol{\pi})$: Optimize the doubly robust estimator [7]; i.e., $\lambda(x,a) = \beta(a|x)^{-1}$ and $\tau > 0$ in (11).
**POEM**: Combines importance corrected empirical risk estimation, $\hat{R}(\boldsymbol{\pi})$, with a regularization that penalizes the variance of the estimated $\hat{R}(\boldsymbol{\pi})$ [34]. We tuned the additional regularization factor $\lambda$. (To keep a fair comparison with the other methods, we did not impose capping on the importance weight here, which was additionally tuned in [34].)
$\mathbf{D_{F^*}}(\boldsymbol{p}\|\boldsymbol{\pi})$: The imputation strategy for the reward uses $\lambda(x,a) = \beta(a|x)^{-1}$ and we tune $\tau > 0$.
**Composite**: We combine the $\hat{R}(\boldsymbol{\pi})$ objective with $\mathbf{D_{F^*}}(\boldsymbol{p}\|\boldsymbol{\pi})$. In addition to the scaling factor $\tau$, we tuned the combination weights.

Table 2 reports the estimated reward obtained by each method on test data. Here we can see that training with the proposed surrogate performs competitively against previous state-of-the-art-methods.

## 4 Conclusion

We investigated alternative objectives for policy optimization in cost-sensitive classification and contextual bandits. The formulations developed are directly applicable to deep learning and improve the underlying optimization landscape. The empirical results in both the cost-sensitive classification and batch contextual bandit scenarios replicate or surpass training with state-of-the-art baseline objectives, merely through the optimization of the non-standard loss functions. There remain several opportunities for further development of surrogate training objectives for sequential decision making tasks (i.e. in planning and reinforcement learning).

## Footnotes

[1] Appendix and code available at `https://www.cs.ualberta.ca/~dale/neurips19/supplement`

[2]The number reported here is lower than reported in the original paper [34]. One hypothesis is that this is due to the removal of the cap on importance weights. A similar result to what we obtain was reported in [20].

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
