[Supplementary Material 1]

# Supplement

# Surrogate Objectives for Batch Policy Optimization in One-step Decision Making

**Minmin Chen**$^*$    **Ramki Gummadi**$^*$    **Chris Harris**$^*$    **Dale Schuurmans**$^{*\dagger}$
$^*$Google                                          $^\dagger$University of Alberta

## 1 Definitions

Throughout this appendix we use the same notation and definitions from the main body of the paper. In particular, for a vector $\boldsymbol{q} \in \mathbb{R}^K$ let

$$
\begin{aligned}
\boldsymbol{\pi}(\boldsymbol{q}) &= \boldsymbol{f}(\boldsymbol{q}) & (1)\\
\boldsymbol{f}(\boldsymbol{q}) &= e^{\boldsymbol{q}-F(\boldsymbol{q})} & (2)\\
F(\boldsymbol{q}) &= \log(\boldsymbol{1}\cdot e^{\boldsymbol{q}}). & (3)
\end{aligned}
$$

We also use the same risk definitions as the main body, in particular:

*local risk*

$$
\begin{aligned}
\mathcal{R}(\boldsymbol{\pi},\boldsymbol{r},x) &= -\boldsymbol{r}\cdot\boldsymbol{\pi}(x) & (4)\\
\mathcal{R}^*(\boldsymbol{r},x) &= \inf_{\boldsymbol{q}\in\mathcal{Q}}\mathcal{R}(\boldsymbol{f}\circ\boldsymbol{q},\boldsymbol{r},x), & (5)
\end{aligned}
$$

*expected risk*

$$
\mathcal{R}(\boldsymbol{\pi}) = -\mathbb{E}[\boldsymbol{\pi}(x)\cdot\boldsymbol{r}], \qquad (6)
$$

*local smoothed risk*

$$
\begin{aligned}
\mathcal{S}_\tau(\boldsymbol{\pi},\boldsymbol{r},x) &= -\boldsymbol{r}\cdot\boldsymbol{\pi}(x)+\tau\boldsymbol{\pi}\cdot\log\boldsymbol{\pi}(x) & (7)\\
\mathcal{S}_\tau^*(\boldsymbol{r},x) &= \inf_{\boldsymbol{q}\mathcal{Q}}\mathcal{S}_\tau(\boldsymbol{f}\circ\boldsymbol{q},\boldsymbol{r},x) & (8)\\
\mathcal{G}_\tau(\boldsymbol{\pi},\boldsymbol{r},x) &= \mathcal{S}_\tau(\boldsymbol{\pi},\boldsymbol{r},x)-\mathcal{S}_\tau^*(\boldsymbol{r},x), & (9)
\end{aligned}
$$

*expected smoothed risk*

$$
\begin{aligned}
\mathcal{S}_\tau(\boldsymbol{\pi}) &= \mathbb{E}[\mathcal{S}_\tau(\boldsymbol{\pi},\boldsymbol{r},x)] & (10)\\
\mathcal{S}_\tau^* &= \inf_{\boldsymbol{q}\mathcal{Q}}\mathcal{S}_\tau(\boldsymbol{f}\circ\boldsymbol{q}) & (11)\\
\mathcal{G}_\tau(\boldsymbol{\pi}) &= \mathcal{S}_\tau(\boldsymbol{\pi})-\mathcal{S}_\tau^*. & (12)
\end{aligned}
$$

## 2 Proofs for Section 2: Cost-sensitive Classification

**Theorem 1** *Even for a single context $x$, a deterministic reward vector $\boldsymbol{r}$, and a linear model $\boldsymbol{q}(x) = W\boldsymbol{\phi}(x)$, the function $\boldsymbol{r}\cdot\boldsymbol{f}(\boldsymbol{q}(x))$ can have a number of local maxima in $W$ that is exponential in the number of actions $K$ and the number of features in $\boldsymbol{\phi}$.*

*Proof:* To demonstrate the possibility of separated local maxima, start by considering a concrete construction with 5 actions and 1 feature. In particular, let

$$\boldsymbol{r}_1 \;=\; \begin{bmatrix} 1 \\ 2 \\ -1 \\ 2 \\ 1 \end{bmatrix} \quad \text{and} \quad \Phi_1 \;=\; \begin{bmatrix} 2 \\ 1 \\ 0 \\ -1 \\ -2 \end{bmatrix} \tag{13}$$

hence $\boldsymbol{q} = \Phi_1 w$ for a scalar parameter $w$. Note that in this case the policy is given by

$$\boldsymbol{\pi} \;=\; \boldsymbol{f}(\Phi_1 w) \;=\; \frac{1}{d(w)} \begin{bmatrix} e^{2w} \\ e^{w} \\ 1 \\ e^{-w} \\ e^{-2w} \end{bmatrix}, \tag{14}$$

$$\text{where} \quad d(w) \;=\; e^{2w} + e^{w} + 1 + e^{-w} + e^{-2w} \;=\; 2\cosh(2w) + 2\cosh(w) + 1. \tag{15}$$

Therefore, the value function is given by

$$v(w) \;=\; \boldsymbol{r}_1^\top \boldsymbol{\pi} \;=\; \frac{2\cosh(2w) + 4\cosh(w) - 1}{2\cosh(2w) + 2\cosh(w) + 1} \;=\; \frac{n(w)}{d(w)}, \tag{16}$$

$$\text{where} \quad n(w) \;=\; 2\cosh(2w) + 4\cosh(w) - 1. \tag{17}$$

To determine the critical points of the value function, consider the derivative

$$\frac{dv}{dw} \;=\; \frac{2\sinh(w)(8\cosh(w) - 4\cosh^2(w) + 1)}{d(w)^2}. \tag{18}$$

Recall that $\cosh(w) \geq 1$, hence $d(w) \geq 5$, and therefore the zeros for $\frac{dv}{dw}$ occur whenever the numerator is zero. This implies there are exactly three critical points, at $w = 0$ and $w = \pm\operatorname{acosh}(1 + \frac{\sqrt{5}}{2})$ ($\approx \pm 1.3826$). One can also determine that $n(w) \geq d(w)$, hence $v(w) \geq 1$. Finally, observe that since $\cosh(w)$ is an even function, so must be $n(w)$, $d(w)$, and $v(w)$. The function $v(w)$ is plotted in Figure 1.

Figure 1: Plot of the value function $v(w)$.

We can now use this simple example as a widget for creating a combinatorial explosion of local maxima. We achieve this by tiling the previous construction as follows. Let $t$ denote the number of tiles. Expand the previous construction to $5t$ actions and $t$ features by replicating $\boldsymbol{r}_1$ and $\Phi_1$, each $t$ times, in the following manner:

$$\boldsymbol{r}_t \;=\; \boldsymbol{1} \otimes \boldsymbol{r}_1 \;=\; \begin{bmatrix} \boldsymbol{r}_1 \\ \vdots \\ \boldsymbol{r}_1 \end{bmatrix} \quad \text{and} \quad \Phi_t \;=\; I \otimes \Phi_1 \;=\; \begin{bmatrix} \Phi_1 & & \\ & \ddots & \\ & & \Phi_1 \end{bmatrix}, \tag{19}$$

hence $\boldsymbol{r}_t$ is a $5t \times 1$ vector and $\Phi_t$ is a $5t \times t$ matrix. A policy over $5t$ actions can then be parameterized by a $t$ dimensional weight vector $\boldsymbol{w}$ via

$$\boldsymbol{\pi} \;=\; \boldsymbol{f}(\Phi_t \boldsymbol{w}) \;=\; \frac{1}{d(\boldsymbol{w})} \begin{bmatrix} e^{2w_1} \\ e^{w_1} \\ 1 \\ e^{-w_1} \\ e^{-2w_1} \\ \vdots \\ e^{2w_t} \\ e^{w_t} \\ 1 \\ e^{-w_t} \\ e^{-2w_t} \end{bmatrix}, \quad \text{where} \quad d(\boldsymbol{w}) \;=\; \sum_{i=1}^{t} d(w_i). \qquad (20)$$

The value function in this case then becomes

$$v(\boldsymbol{w}) \;=\; \boldsymbol{r}_t^{\top} \boldsymbol{\pi} \;=\; \frac{\sum_{i=1}^{t} n(w_i)}{\sum_{i=1}^{t} d(w_i)}, \quad \text{where} \quad n(\boldsymbol{w}) = \sum_{i=1}^{t} n(w_i). \qquad (21)$$

To determine the locations of the critical points, consider the partial derivative of $v$ with respect to a single parameter, say $w_i$:

$$\frac{\partial v}{\partial w_i} \;=\; \frac{n'(w_i)d(\boldsymbol{w}) - d'(w_i)n(\boldsymbol{w})}{d(\boldsymbol{w})^2} \;=\; \frac{n'(w_i) - d'(w_i)v(\boldsymbol{w})}{d(\boldsymbol{w})}. \qquad (22)$$

As before, since $d(w_i) \geq 1$ for all $i$, hence $d(\boldsymbol{w}) \geq t$, we know the zeros of $\frac{\partial v}{\partial w_i}$ are determined by $w_i$ such that $n'(w_i) = d'(w_i)v(\boldsymbol{w})$. One root value for $w_i$ will always be $w_i = 0$, regardless of the other values of $w_j$, $j \neq i$, since the individual numerator and denominator functions each satisfy $n'(0) = d'(0) = 0$ respectively. It remains only to show that there are always two other roots for $w_i$, symmetrically placed around but separated from 0, regardless of the values for the other $w_j$, $j \neq i$.

A few useful properties of $v(\boldsymbol{w})$ will allow us to show this. First, since the individual $n(w_i)$ and $d(w_i)$ functions are even, the function $v(\boldsymbol{w})$ must also be even along any coordinate $w_i$. Second, since $n(w_i) \geq d(w_i)$ for all $i$, and moreover $n(w_i) > d(w_i)$ if $w_i \neq 0$, we have $v(\boldsymbol{w}) > 1$ if $w_i \neq 0$. Third, since it always holds that $n(w_i) < 2d(w_i)$, we also have $\sum_i n(w_i) < 2 \sum_i d(w_i)$, hence $v(\boldsymbol{w}) < 2$. Therefore, even though the value of $v(\boldsymbol{w})$ will determine the exact location of the symmetric nonzero roots in (22), we can establish the existence of these nonzero roots simply by assuming $v(\boldsymbol{w})$ takes on any arbitrary value in the range $1 < v < 2$, as we now show.

Consider the zeros of $n'(w_i) - d'(w_i)v$ where $v$ is any quantity such that $1 < v < 2$. From the definitions we know that $n'(w_i) = 4\sinh(2w_i) + 4\sinh(w_i)$ and $d'(w_i) = 4\sinh(2w_i) + 2\sinh(w_i)$, hence we seek the values of $w_i$ such that

$$2(1-v)\sinh(2w_i) \;=\; (v-2)\sinh(w_i) \qquad (23)$$

As noted, one solution is $w_i = 0$ but we particularly seek the nonzero roots, so consider $w_i \neq 0$, hence $\sinh(w_i) \neq 0$. Under this assumption (23) reduces to

$$4(1-v)\cosh(w_i) \;=\; v-2, \qquad (24)$$

which has a putative solution pair $w_i = \pm \operatorname{acosh}(\frac{2-v}{4(v-1)})$. This solution pair exists (and is nonzero) if $1 < \frac{2-v}{4(v-1)} < \infty$, which is guaranteed by $1 < v < 2$.

To summarize: in characterizing the landscape of $v(\boldsymbol{w})$, we know the function is continuous, smooth, and sandwiched between $1 \leq v(\boldsymbol{w}) < 2$ for all $\boldsymbol{w}$. Along each coordinate axis, $w_i$, regardless of the values of the other weight parameters, $w_j$, $j \neq i$, there are exactly three critical points: one at zero, and two others symmetrically placed around but distinct from zero (attaining equal value, since $v$ is an even function along each coordinate). Since the point $w_i = 0$ is a local minimum along the coordinate, the other two critical points must be local maxima. The overall weight vector $\boldsymbol{w}$ is only at a critical point if each of its coordinates are at a critical point. Therefore, in total there are $3^t$ critical points, of which $2^t$ are local maxima (i.e. each coordinate is at a local maximum).

∎

**Comment**  Clearly, the above construction creates $2^t$ local maxima that all have the same expected value. Intuitively, a small perturbation of one of the modes in the initial construction can preserve the number of local maxima while elevating a single such maximum to global dominance.

**Proposition 2**  *Let $\tilde{\boldsymbol{\pi}}_\tau = \arg\min_{\boldsymbol{\pi}\in\mathcal{P}} \mathcal{S}_\tau(\boldsymbol{\pi})$. Then $\tilde{\boldsymbol{\pi}}_\tau(x) = \exp(\mathbb{E}[\boldsymbol{r}|x] - F(\mathbb{E}[\boldsymbol{r}|x])/\tau)$ and $\mathcal{R}(\tilde{\boldsymbol{\pi}}_\tau) < \mathcal{R}^* + \tau\log K$. Hence for any $\epsilon > 0$ setting $\tau < \epsilon/\log K$ ensures $\mathcal{R}(\tilde{\boldsymbol{\pi}}_\tau) < \mathcal{R}^* + \epsilon$.*

*Proof:*  Let $\Delta(\boldsymbol{z})$ denote putting a vector $\boldsymbol{z}$ on the main diagonal of a square matrix. First, it is easy to prove that $\tilde{\boldsymbol{\pi}}_\tau(x) = \exp(\mathbb{E}[\boldsymbol{r}|x] - F(\mathbb{E}[\boldsymbol{r}|x])/\tau)$ is optimal. Consider a fixed $x$ and note:

$$\frac{d\mathcal{S}_\tau(\boldsymbol{\pi},\boldsymbol{r},x)}{d\boldsymbol{q}(x)} = \left(\Delta(\boldsymbol{\pi}(x)) - \boldsymbol{\pi}(x)\boldsymbol{\pi}(x)^\top\right)(\tau\boldsymbol{q} - \boldsymbol{r}), \tag{25}$$

thus, $\boldsymbol{q}(x) = \boldsymbol{r}/\tau - \mathbf{1}v/\tau$ determines an equilibrium point in $\mathcal{S}_\tau(\boldsymbol{f}\circ\boldsymbol{q},\boldsymbol{r},x)$ for any constant $v$. Since (25) is linear in $\boldsymbol{r}$, taking an expectation in $\boldsymbol{r}$ still yields equilibria of the form $\boldsymbol{q}(x) = \mathbb{E}[\boldsymbol{r}/\tau|x]$ setting $v = 0$. Thus, the optimal policy conditioned on $x$ can be written as $\tilde{\boldsymbol{\pi}}_\tau(x) = \exp\left(\mathbb{E}[\boldsymbol{r}|x]/\tau - F(\mathbb{E}[\boldsymbol{r}|x]/\tau)\right) = \exp\left(\bar{\boldsymbol{r}}/\tau - F(\bar{\boldsymbol{r}}/\tau)\right)$, where we let $\bar{\boldsymbol{r}} = \mathbb{E}[\boldsymbol{r}|x]$.

For the second part of the claim, for any fixed $x$ and $\boldsymbol{r}$ we consider the gap between the exact optimum and the approximate optimum produced by $\tilde{\boldsymbol{\pi}}_\tau(x) = \boldsymbol{f}(\boldsymbol{q}(x)) = \boldsymbol{f}(\frac{\bar{\boldsymbol{r}}}{\tau})$:

$$\text{Gap} = \max_a r_a - \boldsymbol{f}(\tfrac{\bar{\boldsymbol{r}}}{\tau})\cdot\bar{\boldsymbol{r}}. \tag{26}$$

We can bound this gap by lower bounding the expected reward achieved by the policy at $x$:

$$\boldsymbol{f}(\tfrac{\bar{\boldsymbol{r}}}{\tau})\cdot\bar{\boldsymbol{r}} = \tau\boldsymbol{f}(\tfrac{\bar{\boldsymbol{r}}}{\tau})\cdot\tfrac{\bar{\boldsymbol{r}}}{\tau} \tag{27}$$

$$= \tau F(\tfrac{\bar{\boldsymbol{r}}}{\tau}) + \tau F^*(\boldsymbol{f}(\tfrac{\bar{\boldsymbol{r}}}{\tau})) \tag{28}$$

$$\geq \max_a r_a - \tau\log K, \tag{29}$$

where the second step uses the fact that the Young-Fenchel inequality is tight at a dual pair $\tilde{\boldsymbol{\pi}}$ and $\boldsymbol{q}$ [2, §3.3.2], and the last step using the fact that $F^*$ is negative entropy and the maximum entropy of any distribution over $K$ actions is $\log K$. From this it is easy to conclude that whenever $\tau \leq \epsilon/\log K$ we must have Gap $\leq \epsilon$. The result then follows by noting that this inequality holds pointwise for all $x$, hence also in expectation over $x$. ∎

**Theorem 3**  *For an arbitrary baseline $v$ and $\tau > 0$, let*

$$L(\boldsymbol{q},\boldsymbol{r},x) = \tau D_F\left(\boldsymbol{q}(x) + \tfrac{v}{\tau}\,\Big\|\,\tfrac{\boldsymbol{r}}{\tau}\right) + \tfrac{\tau}{4}\left\|\boldsymbol{q}(x) - \tfrac{\boldsymbol{r}-v}{\tau}\right\|^2, \tag{30}$$

*Then, for any fixed $v$, $L$ is strongly convex in $\boldsymbol{q}$ and calibrated with respect to the smoothed (shifted) risk $\mathcal{S}_\tau(\boldsymbol{f}\circ\boldsymbol{q},\boldsymbol{r} - v, x) = \mathcal{S}_\tau(\boldsymbol{f}\circ\boldsymbol{q},\boldsymbol{r},x) - v$ with calibration function $\delta(\epsilon, x) = \epsilon\ \forall x$.*

*Proof:*  Strong convexity is immediate from the inclusion of the squared loss. We need to establish two additional properties. First, that the global minimizer of (30) is also a global minimizer of the local smoothed risk $\mathcal{S}_\tau(\boldsymbol{f}\circ\boldsymbol{q},\boldsymbol{r}-v,x)$ (7). Second, that the surrogate objective is an *upper bound* on the suboptimality of the local smoothed risk.

For the equilibrium condition, note that $\mathcal{S}_\tau(\boldsymbol{f}\circ\boldsymbol{q},\boldsymbol{r}-v,x)$ must satisfy (25), hence, again, we have $\boldsymbol{q}(x) = \boldsymbol{r}/\tau - \mathbf{1}v/\tau$ is an equilibrium point for any fixed $v$. By comparison, taking the gradient of the surrogate with respect to $\boldsymbol{q}(x)$ yields

$$\frac{dL(\boldsymbol{q},\boldsymbol{r},x)}{d\boldsymbol{q}(x)} = \tau(\tilde{\boldsymbol{\pi}}(x) - \boldsymbol{p}(x)) + \tfrac{\tau}{2}\left(\boldsymbol{q}(x) - \tfrac{\boldsymbol{r}}{\tau} + \tfrac{v}{\tau}\right), \tag{31}$$

where $\tilde{\boldsymbol{\pi}}(x) = \boldsymbol{f}(\boldsymbol{q}(x) + \tfrac{v}{\tau})$ for any fixed $v$ and $\boldsymbol{p} = \boldsymbol{f}(\tfrac{\boldsymbol{r}}{\tau})$. Thus, an equilibrium point for $L(\boldsymbol{q},\boldsymbol{r},x)$ is also given by $\boldsymbol{q}(x) = \tfrac{\boldsymbol{r}}{\tau} - \tfrac{v}{\tau}$. Moreover, any such point must be a *unique* global minimizer for $L(\boldsymbol{q},\boldsymbol{r},x)$ by strong convexity. Since this choice of $\boldsymbol{q}(x)$ uniquely determines $\boldsymbol{\pi}(x)$, it characterizes the equilibria of $\mathcal{S}(\boldsymbol{\pi},\boldsymbol{r}-v,x)$ and therefore also the global minimizer.

For the second part, first let $\mathcal{S}^*(\boldsymbol{r}-v,x) = \inf_{\boldsymbol{q}\in\mathcal{Q}}\mathcal{S}(\boldsymbol{f}\circ\boldsymbol{q},\boldsymbol{r}-v,x)$, and note that since we know

$$\mathcal{S}(\boldsymbol{f}\circ\boldsymbol{q},\boldsymbol{r}-v,x) = -\tau F(\tfrac{\boldsymbol{r}-v}{\tau}) + \tau D_F\left(\tfrac{\boldsymbol{r}-v}{\tau}\,\Big\|\,\boldsymbol{q}(x)\right) \tag{32}$$

$$= v - \tau F(\tfrac{\boldsymbol{r}}{\tau}) + \tau D_F\left(\tfrac{\boldsymbol{r}-v}{\tau}\,\Big\|\,\boldsymbol{q}(x)\right), \tag{33}$$

it follows that $S^*(\boldsymbol{r} - v, x) = v - \tau F(\frac{\boldsymbol{r}}{\tau})$, which is achieved at $\boldsymbol{q} = \boldsymbol{r}/\tau - \mathbf{1}v/\tau$. Finally, to establish that $L(\boldsymbol{q}, \boldsymbol{r}, x) \geq S(\boldsymbol{f} \circ \boldsymbol{q}, \boldsymbol{r} - v, x) - S^*(\boldsymbol{r} - v, x)$ we consider a second order Taylor analysis along the lines of [6], which uses two Taylor expansions of $F(\boldsymbol{q})$. Using the same derivation, it can be shown that

$$
\begin{aligned}
D_F\left(\tfrac{\boldsymbol{r}}{\tau}\middle\|\boldsymbol{q}(x) + \tfrac{v}{\tau}\right) &= D_F\left(\boldsymbol{q}(x) + \tfrac{v}{\tau}\middle\|\tfrac{\boldsymbol{r}}{\tau}\right) \\
&\quad + \frac{1}{4}\left(\boldsymbol{q}(x) - \tfrac{\boldsymbol{r}}{\tau} + \tfrac{v}{\tau}\right)^\top (H_F(\boldsymbol{b}) - H_F(\boldsymbol{a}))\left(\boldsymbol{q}(x) - \tfrac{\boldsymbol{r}}{\tau} + \tfrac{v}{\tau}\right), \quad (34)
\end{aligned}
$$

where $H_F$ denotes the Hessian of $F$, $\boldsymbol{a} = (1 - \frac{\eta}{2})\frac{\boldsymbol{r}}{\tau} + \frac{\eta}{2}(\boldsymbol{q}(x) + \frac{v}{\tau})$ for some $0 \leq \eta \leq 1$, and $\boldsymbol{b} = (1 - \frac{\rho}{2})(\boldsymbol{q}(x) + \frac{v}{\tau}) + \frac{\rho}{2}\frac{\boldsymbol{r}}{\tau}$ for some $0 \leq \rho \leq 1$. Since the Hessian has the form

$$
H_F(\boldsymbol{a}) = \Delta(\boldsymbol{f}(\boldsymbol{a})) - \boldsymbol{f}(\boldsymbol{a})\boldsymbol{f}(\boldsymbol{a})^\top \tag{35}
$$

for all $\boldsymbol{a}$, we know that $I \succeq H_F(\boldsymbol{a}) \succeq 0$ and $I \succeq H_F(\boldsymbol{b}) \succeq 0$, hence $I \succeq H_F(\boldsymbol{b}) - H_F(\boldsymbol{a})$. Therefore, from (34) it follows that

$$
D_F\left(\tfrac{\boldsymbol{r}}{\tau}\middle\|\boldsymbol{q}(x) + \tfrac{v}{\tau}\right) \leq D_F\left(\boldsymbol{q}(x) + \tfrac{v}{\tau}\middle\|\tfrac{\boldsymbol{r}}{\tau}\right) + \tfrac{1}{4}\left\|\boldsymbol{q}(x) - \tfrac{\boldsymbol{r}}{\tau} + \tfrac{v}{\tau}\right\|^2. \tag{36}
$$

Therefore,

$$
\begin{aligned}
S(\boldsymbol{f} \circ \boldsymbol{q}, \boldsymbol{r} - v, x) &= \tau D_F\left(\tfrac{\boldsymbol{r}}{\tau}\middle\|\boldsymbol{q}(x) + \tfrac{v}{\tau}\right) + v - \tau F\left(\tfrac{\boldsymbol{r}}{\tau}\right) & (37) \\
&\leq \tau D_F\left(\boldsymbol{q}(x) + \tfrac{v}{\tau}\middle\|\tfrac{\boldsymbol{r}}{\tau}\right) + \tfrac{\tau}{4}\left\|\boldsymbol{q}(x) - \tfrac{\boldsymbol{r}}{\tau} + \tfrac{v}{\tau}\right\|^2 + v - \tau F\left(\tfrac{\boldsymbol{r}}{\tau}\right) & (38) \\
&= L(\boldsymbol{q}, \boldsymbol{r}, c) + v - \tau F\left(\tfrac{\boldsymbol{r}}{\tau}\right). & (39)
\end{aligned}
$$

We conclude that if $L(\boldsymbol{q}, \boldsymbol{r}, x) \leq \epsilon$ then $S(\boldsymbol{f} \circ \boldsymbol{q}, \boldsymbol{r} - v, x) - S^*(\boldsymbol{r} - v, x) \leq \epsilon$ and the result follows. ■

## 3 Proofs for Section 3: Batch Contextual Bandits

Note that throughout this section, as in the main body of the paper, we use $\hat{\boldsymbol{r}}(x)$ to denote the imputed reward estimator

$$
\hat{\boldsymbol{r}}(x) = \tau\boldsymbol{q}(x) + \mathbf{1}_a\lambda(x, a)(r_a - \tau\boldsymbol{q}(x)_a). \tag{40}
$$

This simplified notation allows us to simply write $\hat{\boldsymbol{r}}$ in place of $\boldsymbol{r}$ in the expressions below. However, this notation also masks the dependence of $\hat{\boldsymbol{r}}(x)$ on the model output $\boldsymbol{q}$ and the observation $(x, a, r_a)$. That is, to be more explicit, the full dependence of $\hat{\boldsymbol{r}}$ can be fully expressed as $\hat{\boldsymbol{r}}(x, a, r_a, \boldsymbol{q}(x))$.

**Proposition 4** *For any $\boldsymbol{q}$, $\tau > 0$ and observation $(x, a, r_a)$:* $\tau D_F\left(\frac{\hat{\boldsymbol{r}}(x)}{\tau}\middle\|\boldsymbol{q}(x)\right) = \mathcal{G}_\tau(\boldsymbol{f} \circ \boldsymbol{q}, \hat{\boldsymbol{r}}, x)$.

*Proof:* By Lemma 13 below we have $S_\tau(\boldsymbol{f} \circ \boldsymbol{q}, \hat{\boldsymbol{r}}, x) = -\tau F\left(\frac{\hat{\boldsymbol{r}}}{\tau}\right) + \tau D_F\left(\frac{\hat{\boldsymbol{r}}}{\tau}\middle\|\boldsymbol{q}(x)\right)$. By Lemma 14 below we also know $S_\tau^*(\hat{\boldsymbol{r}}, x) = -\tau F\left(\frac{\hat{\boldsymbol{r}}}{\tau}\right)$. Hence

$$
\begin{aligned}
\mathcal{G}_\tau(\boldsymbol{f} \circ \boldsymbol{q}, \hat{\boldsymbol{r}}, x) &= S_\tau(\boldsymbol{f} \circ \boldsymbol{q}, \hat{\boldsymbol{r}}, x) - S_\tau^*(\hat{\boldsymbol{r}}, x) & (41) \\
&= \left(-\tau F\left(\tfrac{\hat{\boldsymbol{r}}}{\tau}\right) + \tau D_F\left(\tfrac{\hat{\boldsymbol{r}}}{\tau}\middle\|\boldsymbol{q}(x)\right)\right) - \left(-\tau F\left(\tfrac{\hat{\boldsymbol{r}}}{\tau}\right)\right) & (42) \\
&= \tau D_F\left(\tfrac{\hat{\boldsymbol{r}}}{\tau}\middle\|\boldsymbol{q}(x)\right). & (43)
\end{aligned}
$$

■

**Theorem 5** *For any model $\boldsymbol{q}$, $\tau > 0$, observation $(x, a, r_a)$, and baseline $v$:*

$$
L(\boldsymbol{q}, \hat{\boldsymbol{r}}, x) \geq \tau D_F\left(\tfrac{\hat{\boldsymbol{r}}(x)}{\tau}\middle\|\boldsymbol{q}(x) + \tfrac{v}{\tau}\right) = \mathcal{G}_\tau(\boldsymbol{f} \circ \boldsymbol{q}, \hat{\boldsymbol{r}}, x) \geq 0. \tag{44}
$$

*Moreover, $L$ is calibrated with respect to $S_\tau(\boldsymbol{f} \circ \boldsymbol{q}, \hat{\boldsymbol{r}} - v, x)$ with calibration function $\delta(x, \epsilon) = \epsilon$.*

*Proof:* The middle equality in (44) is established by Proposition 4 combined with the shift invariance of $D_F$ established in Lemma 12 below. The last inequality in (44) follows immediately from the definition of $\mathcal{G}_\tau$. The first inequality in (44) follows from the definition $L(\boldsymbol{q}, \hat{\boldsymbol{r}}, x) =$

$\tau D_F\left(\boldsymbol{q}(x) + \frac{v}{\tau} \middle\| \frac{\hat{r}}{\tau}\right) + \frac{\tau}{4}\left\|\boldsymbol{q}(x) - \frac{\hat{r}-v}{\tau}\right\|^2$ combined with the inequality (36) established in the proof of Theorem 3.

Finally, note that $L$ is also nonnegative, yet $L(\boldsymbol{q}, \hat{\boldsymbol{r}}, x) = 0$ at $\boldsymbol{q}(x) = \frac{\hat{r}-v}{\tau}$, which implies this is a global minimizer of $L$, which also must achieve suboptimality gap $\mathcal{G}_\tau(\boldsymbol{f} \circ \boldsymbol{q}, \hat{\boldsymbol{r}} - v, x) = 0$ since $L$ dominates $\mathcal{G}_\tau$. Hence, any desired upper bound $\epsilon > 0$ on the suboptimality $\mathcal{G}_\tau(\boldsymbol{f} \circ \boldsymbol{q}, \hat{\boldsymbol{r}} - v, x)$ is achieved by finding a $\boldsymbol{q}$ such that $L(\boldsymbol{q}, \hat{\boldsymbol{r}}, x) \leq \epsilon$. ∎

**Theorem 6** *For any model $\boldsymbol{q}$, any $\hat{\boldsymbol{r}}$ such that $\mathbb{E}[\hat{\boldsymbol{r}}|x] = \mathbb{E}[\boldsymbol{r}|x]$, and any baseline $v$:*

$$\mathbb{E}[L(\boldsymbol{q}, \hat{\boldsymbol{r}}, x)] \;\;\geq\;\; \mathbb{E}\left[\tau D_F\left(\frac{\hat{r}(x)}{\tau} \middle\| \boldsymbol{q}(x) + \frac{v}{\tau}\right)\right] \;\;\geq\;\; \mathcal{G}_\tau(\boldsymbol{f} \circ \boldsymbol{q}) \;\;\geq\;\; 0. \tag{45}$$

*Proof:* Assume a fixed $\boldsymbol{q}$, and note that $\hat{r}(x)$ is a random vector derived from $\boldsymbol{q}$ and the sample $(x, a, r_a) \sim p(x, \boldsymbol{r})\beta(a|x)$ (i.e., $a$ is independent of $\boldsymbol{r}$ given $x$). The last inequality in (45) is immediate from the definition of $\mathcal{G}_\tau(\boldsymbol{f} \circ \boldsymbol{q})$. The first inequality in (45) is also immediate given Theorem 5, which establishes $L(\boldsymbol{q}, \hat{\boldsymbol{r}}, x) \geq \tau D_F\left(\frac{\hat{r}(x)}{\tau} \middle\| \boldsymbol{q}(x) + \frac{v}{\tau}\right)$ pointwise for all observations $(x, a, r_a)$. To establish the middle inequality in (45), first note that for every fixed $x$, the function $\mathcal{S}_\tau^*(\boldsymbol{r}, x) = \inf_{\boldsymbol{q} \in \mathcal{Q}} S(\boldsymbol{f} \circ \boldsymbol{q}, \boldsymbol{r}, x)$ is a pointwise infemum of linear functions of $\boldsymbol{r}$, hence concave in $\boldsymbol{r}$ [2, §3.2.3]. Thus we obtain

$$\mathbb{E}\left[\tau D_F\left(\frac{\hat{r}(x)}{\tau} \middle\| \boldsymbol{q}(x) + \frac{v}{\tau}\right)\right]$$

$$= \;\; \mathbb{E}\left[\tau D_F\left(\frac{\hat{r}(x)}{\tau} \middle\| \boldsymbol{q}(x)\right)\right] \quad \text{by Lemma 12 below} \tag{46}$$

$$= \;\; \mathbb{E}\left[\mathcal{S}_\tau(\boldsymbol{f} \circ \boldsymbol{q}, \hat{\boldsymbol{r}}, x) - \mathcal{S}_\tau^*(\hat{\boldsymbol{r}}, x)\right] \quad \text{by Proposition 4} \tag{47}$$

$$= \;\; \mathbb{E}\left[\mathcal{S}_\tau(\boldsymbol{f} \circ \boldsymbol{q}, \hat{\boldsymbol{r}}, x)\right] - \mathbb{E}\left[\inf_{\boldsymbol{q} \in \mathcal{Q}} \mathcal{S}_\tau(\boldsymbol{f} \circ \boldsymbol{q}, \hat{\boldsymbol{r}}, x)\right] \tag{48}$$

$$\geq \;\; \mathbb{E}\left[\mathcal{S}_\tau(\boldsymbol{f} \circ \boldsymbol{q}, \hat{\boldsymbol{r}}, x)\right] - \inf_{\boldsymbol{q} \in \mathcal{Q}} \mathbb{E}\left[\mathcal{S}_\tau(\boldsymbol{f} \circ \boldsymbol{q}, \hat{\boldsymbol{r}}, x)\right] \quad \text{by Jensen's inequality} \tag{49}$$

$$= \;\; \mathbb{E}\left[\mathcal{S}_\tau(\boldsymbol{f} \circ \boldsymbol{q}, \boldsymbol{r}, x)\right] - \inf_{\boldsymbol{q} \in \mathcal{Q}} \mathbb{E}\left[\mathcal{S}_\tau(\boldsymbol{f} \circ \boldsymbol{q}, \boldsymbol{r}, x)\right] \tag{50}$$

$$\text{by linearity of } \mathcal{S}_\tau \text{ with respect to } \boldsymbol{r} \text{ and unbiasedness of } \hat{\boldsymbol{r}}$$

$$= \;\; \mathcal{S}_\tau(\boldsymbol{f} \circ \boldsymbol{q}) - \mathcal{S}_\tau^* \;\; = \;\; \mathcal{G}_\tau(\boldsymbol{f} \circ \boldsymbol{g}). \tag{51}$$

∎

## 3.1 Concentration

To keep the technical presentation straightforward, we assume the domain $X$ is a bounded subset of $\mathbb{R}^n$ for some $n$.

### 3.1.1 Well-behavedness conditions for concentration

For concentration to hold uniformly over a class of random variables, such as those defined by scalar-valued divergences between model outputs $\boldsymbol{q}(x)$ and estimated rewards $\hat{r}(x)$, we need to impose a set of assumptions to ensure the needed quantities remain appropriately bounded. In particular, we need to assume the following about $p(x, \boldsymbol{r})$, $\beta$, $\hat{\boldsymbol{r}}$ and $\mathcal{H}$:

- There exist constants $c_X$ and $c_R$ such that $\|x\|_2 \leq c_X$ and $\|\boldsymbol{r}\|_\infty \leq c_R$ for all $(x, \boldsymbol{r})$ in the support of $p(x, \boldsymbol{r})$.

- There exists a constant $\rho > 0$ such that $\beta(a|x) \geq \rho$ for all $x \in X$ and $a \in A$.

- $\mathbb{E}[\hat{r}(x)|x] = \mathbb{E}[\boldsymbol{r}|x]$ for all $x$; i.e., $\hat{r}(x)$ is unbiased.

- Every $\boldsymbol{q} \in \mathcal{H}$ can be expressed as a composition of a bounded linear with a general bounded function; that is, $\mathcal{H} = \mathcal{W} \circ \mathcal{Z}$, where $\mathcal{W} = \{W : \|W\|_2 \leq c_W\}$ and $\mathcal{Z} = \{\boldsymbol{z} : \|\boldsymbol{z}(x)\|_2 \leq c_Z \;\forall x \in \text{support}(p(x, \boldsymbol{r}))\}$. This implies $\boldsymbol{q}$ can be expressed as $\boldsymbol{q}(x) = W\boldsymbol{z}(x)$. An example is a neural network with bounded weights; see Section 3.1.3. Let $c_\mathcal{H} = c_W c_Z$.

We say that the collection $p(x, \boldsymbol{r})$, $\beta$, $\hat{\boldsymbol{r}}$ and $\mathcal{H}$ is "well behaved" if the above assumptions are satisfied.

The main consequence of these assumptions is that the Rademacher complexity of the class of random variables of interest will then exhibit reasonable contraction. In particular, we are interested in the scalar valued divergence $D_F\left(\frac{\hat{r}(x)}{\tau}\big\|\boldsymbol{q}(x)\right)$ obtained by a function $\boldsymbol{q} \in \mathcal{H}$ on a given sample $(x, a, r_a)$. Consider the class of scalar-valued functions induced by composing the divergence of interest with a model $\boldsymbol{q} \in \mathcal{H}$:

$$\mathcal{F} = \left\{ d_{a,\boldsymbol{r}} : d_{a,\boldsymbol{r}}(\boldsymbol{q}(x)) = D_F\left(\tfrac{\hat{r}(x)}{\tau}\big\|\boldsymbol{q}(x)\right) \text{ where } \boldsymbol{q} \in \mathcal{H} \right\}, \tag{52}$$

where $a \in A$ and $\boldsymbol{r} \in \operatorname{support}(p(x, \boldsymbol{r}))$. The Rademacher complexity of $\mathcal{F}$ can then be defined as

$$R_T(\mathcal{F}) = \frac{1}{T}\mathbb{E}\left[\sup_{\boldsymbol{q}\in\mathcal{H}} \sum_{i=1}^{T} \sigma_i d_{a_i, \boldsymbol{r}_i}(\boldsymbol{q}(x_i))\right], \tag{53}$$

where the $\sigma_i$ are independent and uniformly distributed over $\{1, -1\}$ [1, 7].

The key to the well-behavedness conditions is that they allow us to establish in Lemma 11 below that there exists a constant $c_\mathcal{F}$ such that

$$R_T(\mathcal{F}) \leq \frac{c_\mathcal{F}}{\sqrt{T}}. \tag{54}$$

### 3.1.2  Main concentration results

Recall the definitions of the empirical surrogate loss and empirical divergence respectively

$$\hat{L}(\boldsymbol{q}, \mathcal{D}) = \frac{1}{T} \sum_{(x_i, a_i, r_i, \beta_i)\in\mathcal{D}} L(\boldsymbol{q}, \hat{\boldsymbol{r}}, x_i) \tag{55}$$

$$\hat{D}(\boldsymbol{q}, \mathcal{D}) = \frac{1}{T} \sum_{(x_i, a_i, r_i, \beta_i)\in\mathcal{D}} D_F\left(\tfrac{\hat{r}(x_i)}{\tau}\big\|\boldsymbol{q}(x_i)\right). \tag{56}$$

**Lemma 7** *Assume $\mathcal{H}$, $\beta$, $p(x, \boldsymbol{r})$ and $\hat{\boldsymbol{r}}$ are "well behaved". Then for any $\tau, \delta > 0$ there exists a constant $C$ such that with probability at least $1 - \delta$:*

$$\mathbb{E}\left[D_F\left(\tfrac{\hat{r}(x)}{\tau}\big\|\boldsymbol{q}(x)\right)\right] \leq \hat{D}_F(\boldsymbol{q}, \mathcal{D}) + \frac{C}{\sqrt{T}} \quad \forall \boldsymbol{q} \in \mathcal{H}. \tag{57}$$

*Proof:*  Assuming well-behavedness, by Lemma 9 below we know that there exists a constant $c_D$ such that $c_D \geq D_F\left(\tfrac{\hat{r}(x)}{\tau}\big\|\boldsymbol{q}(x)\right) \geq 0$ for all $\boldsymbol{q} \in \mathcal{H}$ and $(x, \boldsymbol{r})$ in the support of $p(x, \boldsymbol{r})$. Using this fact, the bound [7, Theorem 26.5] can then be applied to show that with probability at least $1 - \delta$, for all $\boldsymbol{q} \in \mathcal{H}$:

$$\mathbb{E}\left[D_F\left(\tfrac{\hat{r}(x)}{\tau}\big\|\boldsymbol{q}(x)\right)\right] \leq \hat{D}_F(\boldsymbol{q}, \mathcal{D}) + 2R_T(\mathcal{F}) + 4c_D\sqrt{\tfrac{2}{T}\log\tfrac{2}{\delta}}. \tag{58}$$

By Lemma 11 below we also know there exists a constant $c_\mathcal{F}$ such that $R_T(\mathcal{F}) \leq \frac{c_\mathcal{F}}{\sqrt{T}}$, hence $C$ can be chosen to be $2c_\mathcal{F} + 4c_D\sqrt{2\log(2/\delta)}$.  ∎

**Theorem 8** *Assume $\mathcal{H}$, $\beta$, $p(x, \boldsymbol{r})$ and $\hat{\boldsymbol{r}}$ are "well behaved". Then for any $v$ and $\tau, \delta > 0$, there exists a $C$ such that with probability at least $1 - \delta$: if $\hat{L}(\boldsymbol{q}, \mathcal{D}) < \frac{\tau C}{\sqrt{T}}$ for $\boldsymbol{q} \in \mathcal{H}$ then $\mathcal{G}_\tau(\boldsymbol{f} \circ \boldsymbol{q}) \leq \frac{2\tau C}{\sqrt{T}}$.*

*Proof:*  By Theorem 5 we know $L(\boldsymbol{q}, \hat{\boldsymbol{r}}, x) \geq \tau D_F\left(\tfrac{\hat{r}(x)}{\tau}\big\|\boldsymbol{q}(x) + \tfrac{v}{\tau}\right)$ for any $\tau > 0$, model $\boldsymbol{q}$, observation $(x, a, r_a)$, and baseline $v$. Assuming well-behavedness, Lemma 7 above shows that for any $\tau, \delta > 0$ there exists a constant $C$ such that with probability at least $1 - \delta$, for any $\boldsymbol{q} \in \mathcal{H}$:

$$\hat{L}(\boldsymbol{q}, \mathcal{D}) \geq \tau\hat{D}(\boldsymbol{q}, \mathcal{D}) \tag{59}$$

$$\geq \mathbb{E}\left[\tau D_F\left(\tfrac{\hat{r}(x)}{\tau}\big\|\boldsymbol{q}(x)\right)\right] - \tfrac{\tau C}{\sqrt{T}} \tag{60}$$

$$\geq \mathcal{G}_\tau(\boldsymbol{f} \circ \boldsymbol{q}) - \tfrac{\tau C}{\sqrt{T}}, \tag{61}$$

where the last inequality follows from Theorem 6. Assume there is a $\boldsymbol{q} \in \mathcal{H}$ that achieves $\hat{L}(\boldsymbol{q}, \mathcal{D}) \leq \frac{\tau C}{\sqrt{T}}$. Then by (61) it follows that, with probability at least $1 - \delta$:

$$\mathcal{G}_\tau(\boldsymbol{f} \circ \boldsymbol{q}) \leq \hat{L}(\boldsymbol{q}, \mathcal{D}) + \tfrac{\tau C}{\sqrt{T}} \leq \tfrac{2\tau C}{\sqrt{T}}. \tag{62}$$

∎

**Lemma 9** *Assume $\mathcal{H}$, $\beta$, $p(x, \boldsymbol{r})$ and $\hat{\boldsymbol{r}}$ are "well behaved". Then for any $\tau > 0$ there exists a constant $c_D$ such that $c_D \geq D_F\big(\frac{\hat{\boldsymbol{r}}(x)}{\tau}\big\|\boldsymbol{q}(x)\big) \geq 0$ for all $a \in A$, $\boldsymbol{q} \in \mathcal{H}$ and $(x, \boldsymbol{r})$ in the support of $p(x, \boldsymbol{r})$.*

*Proof:* Nonnegativity is immediate. Fix $\tau > 0$, $a \in A$, and recall the definition:

$$D_F\big(\tfrac{\hat{\boldsymbol{r}}(x)}{\tau}\big\|\boldsymbol{q}(x)\big) = F\big(\tfrac{\hat{\boldsymbol{r}}(x)}{\tau}\big) - F\big(\boldsymbol{q}(x)\big) - \boldsymbol{f}(\boldsymbol{q}(x)) \cdot \Big(\tfrac{\hat{\boldsymbol{r}}(x)}{\tau} - \boldsymbol{q}(x)\Big) \tag{63}$$

$$= F\Big(\boldsymbol{q}(x) + \boldsymbol{1}_a \tfrac{r_a/\tau - q(x)_a}{\beta(a|x)}\Big) - F\big(\boldsymbol{q}(x)\big) - \boldsymbol{f}(\boldsymbol{q}(x))_a \tfrac{r_a/\tau - q(x)_a}{\beta(a|x)}. \tag{64}$$

We bound each term. First note that for any $\boldsymbol{q} \in \mathbb{R}^K$ we have $|F(\boldsymbol{q})| \leq \|\boldsymbol{q}\| + \log K$ [2, §3.1.5], hence

$$|F(\boldsymbol{q}(x))| \leq c_\mathcal{H} + \log K \tag{65}$$

$$|F\big(\tfrac{\hat{\boldsymbol{r}}(x)}{\tau}\big)| \leq \|\boldsymbol{q}(x)\| + \big|\tfrac{r_a}{\tau\beta(a|x)}\big| + \big|\tfrac{q(x)_a}{\beta(a|x)}\big| + \log K \tag{66}$$

$$\leq \Big(1 + \tfrac{1}{\rho}\Big) c_\mathcal{H} + \tfrac{c_R}{\tau\rho} + \log K \tag{67}$$

$$\big|\boldsymbol{f}(\boldsymbol{q}(x))_a\big(\tfrac{r_a}{\tau\beta(a|x)} - \tfrac{q(x)_a}{\beta(a|x)}\big)\big| \leq \big|\tfrac{r_a}{\tau\beta(a|x)}\big| + \big|\tfrac{q(x)_a}{\beta(a|x)}\big| \leq \tfrac{c_R}{\tau\rho} + \tfrac{c_\mathcal{H}}{\rho}. \tag{68}$$

Therefore

$$D_F\big(\tfrac{\hat{\boldsymbol{r}}(x)}{\tau}\big\|\boldsymbol{q}(x)\big) \leq 2\Big(1 + \tfrac{1}{\rho}\Big) c_\mathcal{H} + 2\tfrac{c_R}{\tau\rho} + 2\log K. \tag{69}$$

∎

**Lemma 10** *For $\tau > 0$, $\beta \geq \rho$, any $a \in A$ and any $\boldsymbol{r} \in \text{support}(p(x, \boldsymbol{r}))$, the mapping $d_{a,\boldsymbol{r}}(\boldsymbol{q}) = D_F\big(\frac{\hat{\boldsymbol{r}}}{\tau}\big\|\boldsymbol{q}\big)$ is Lipchitz continuous, with Lipschitz bound at most $2\big(1 + \frac{1}{\rho}\big)$.*

*Proof:* For any $a \in A$ and $\boldsymbol{r} \in \text{support}(p(x, \boldsymbol{r}))$, expand the definition as in (64):

$$d_{a,\boldsymbol{r}}(\boldsymbol{q}) = F\Big(\boldsymbol{q} + \boldsymbol{1}_a \tfrac{r_a/\tau - q_a}{\beta_a}\Big) - F(\boldsymbol{q}) - \boldsymbol{f}(\boldsymbol{q})_a \tfrac{r_a/\tau - q_a}{\beta_a}. \tag{70}$$

Note that $\|\nabla F(\boldsymbol{q})\| = \|\boldsymbol{f}(\boldsymbol{q})\| \leq 1$ for all $\boldsymbol{q}$, hence $F(\boldsymbol{q})$ is 1-Lipschitz. A Lipschitz bound can then be formulated for each term in (70), since the mapping $\boldsymbol{q} \mapsto \boldsymbol{q} + \boldsymbol{1}_a \frac{r_a/\tau - q_a}{\beta_a}$ is $\big(1 + \frac{1}{\rho}\big)$-Lipschitz, and the mapping $\boldsymbol{q} \mapsto \boldsymbol{f}(\boldsymbol{q})_a \frac{r_a/\tau - q_a}{\beta_a}$ is $\frac{1}{\rho}$-Lipschitz. Therefore, $d_{a,\boldsymbol{r}}$ is $2\big(1 + \frac{1}{\rho}\big)$-Lipschitz. ∎

**Lemma 11** *Assume $\mathcal{H}$, $\beta$, $p(x, \boldsymbol{r})$ and $\hat{\boldsymbol{r}}$ are "well behaved". Then there exists a constant $c_\mathcal{F}$ such that*

$$R_T(\mathcal{F}) \leq \frac{c_\mathcal{F}}{\sqrt{T}}. \tag{71}$$

*Proof:* To bound the Rademacher complexity of $\mathcal{F}$, it is easier to first consider the Rademacher complexity of $\mathcal{H}$ using the definition for vector-valued functions developed in [5]; define

$$R_T(\mathcal{H}) = \frac{1}{T}\mathbb{E}\left[\sup_{\boldsymbol{q} \in \mathcal{H}} \sum_{i=1}^{T} \sum_{a=1}^{K} \sigma_{ia} \boldsymbol{q}(x_i)_a\right], \tag{72}$$

where the $\sigma_{ij}$ are independent and uniformly distributed over $\{1, -1\}$ [5]. As noted above, $\mathcal{F}$ can then characterized as a composition of the mappings $d_{a,\boldsymbol{r}}(\boldsymbol{q})$ specified in (70) with $\boldsymbol{q} \in \mathcal{H}$. By Lemma 10, we know that each mapping $d_{a,\boldsymbol{r}}$ is Lipschitz continuous with Lipschitz bound at most $\ell_d \stackrel{\triangle}{=} 2\big(1 + \frac{1}{\rho}\big)$. Therefore, the result of [5, Corollary 4] can be applied to establish $R_T(\mathcal{F}) \leq \sqrt{2}\ell_d R_T(\mathcal{H})$.

Then, to bound the Rademacher complexity of $\mathcal{H}$, we exploit the assumed structure $\mathcal{H} = \mathcal{W} \circ \mathcal{Z}$. Here again the result of [5, §4.2] shows that if $\mathcal{H}$ consists of mappings of the form $\boldsymbol{q}(x) = W\boldsymbol{z}(x)$, with $\|W\|_2 \leq c_W$ and $\|\boldsymbol{z}(x)\|_2 \leq c_Z$ for all $x \in \text{support}(p(x, \boldsymbol{r}))$, then $R_T(\mathcal{H}) \leq \frac{\sqrt{2K}\ell_d c_W c_Z}{\sqrt{T}}$. ∎

### 3.1.3 Feedforward neural networks

The well-behavedness conditions are sufficiently general to allow neural network representations for $q(x)$. For example, an $m$-layer feedforward neural network can be written as a composition of matrix multiplications and a nonlinear transfer:

$$q(x) \quad = \quad W^{(m)} \circ \phi \circ W^{(m-1)} \circ \phi \cdots \circ \phi \circ W^{(1)} \circ x, \tag{73}$$

where $W^{(j)}$ are the parameter matrices and $\phi$ is a componentwise transfer with bias:

$$\phi(z) \quad = \quad \begin{bmatrix} \phi(z_1) \\ \vdots \\ \phi(z_K) \\ 1 \end{bmatrix}. \tag{74}$$

Standard choices for $\phi$, such as ReLU, sigmoid and tanh, are Lipschitz bounded. For example, the ReLU transfer $\phi(z) = z_+$ is 1-Lipschitz. This means that if the parameter matrices $W^{(j)}$ are also bounded, i.e., $\|W^{(j)}\|_2 \le B_j$, then $q(x)$ in (73) is itself Lipschitz continuous with Lipschitz constant $B = \prod_{j=1}^{m} B_j$. This can be proved using a straightforward induction [8], exploiting the bounding technique for linear functions in [4].

Consider the class of functions defined by a feedforward neural network (73) with bounded parameters

$$\mathcal{H} \quad = \quad \{q : q(x) = W^{(m)} \circ \phi \cdots \circ \phi \circ W^{(1)} \circ x, \ \|W^{(j)}\|_2 \le B_j, \ \phi \text{ 1-Lipschitz}\}. \tag{75}$$

This class satisfies the well-behavedness conditions for $\mathcal{H}$ stated above, since any $q \in \mathcal{H}$ can be written as $q(x) = W^{(m)} z(x)$ for a function $z(x) = \phi \circ W^{(m-1)} \circ \phi \cdots \circ \phi W^{(1)} \circ x$. Then by construction we have $\|W^{(m)}\| \le B_m \stackrel{\triangle}{=} c_W$ and $\|z(x)\| \le c_X \prod_{j=1}^{m-1} B_j \stackrel{\triangle}{=} c_Z$.

## 3.2 Additional Lemmas

**Lemma 12** *For any $q$, $r$ and scalar $v$:*

$$D_F(q + v\|r) \quad = \quad D_F(q\|r) \tag{76}$$
$$D_F(r\|q + v) \quad = \quad D_F(r\|q); \tag{77}$$

*that is, $D_F$ is shift invariant in either argument.*

*Proof:* First, recall that by the definitions of $f$ and $F$ we have

$$\log f(q + v) \quad = \quad q + v - F(q + v) \quad = \quad q + v - F(q) + v \quad = \quad q - F(q) \quad = \quad \log f(q) \tag{78}$$
$$f(q + v) \quad = \quad f(q) \tag{79}$$
$$F(q + v) \quad = \quad \log \mathbf{1} \cdot e^{q+v} \quad = \quad v + \log \mathbf{1} \cdot e^{q} \quad = \quad F(q) + v. \tag{80}$$

Therefore, for the second identity (77), these identities yield

$$D_F(r\|q + v) \quad = \quad F(r) - r \cdot f(q + v) + F^*(f(q + v)) \tag{81}$$
$$= \quad F(r) - r \cdot f(q) + F^*(f(q)) \tag{82}$$
$$= \quad D_F(r\|q). \tag{83}$$

For the first identity (76), note that $f(r)$ is a probability vector for any $r$, hence

$$D_F(q + v\|r) \quad = \quad F(q + v) - (q + v) \cdot f(r) + F^*(f(r)) \tag{84}$$
$$= \quad F(q) + v - (q \cdot f(r) + v) + F^*(f(r)) \tag{85}$$
$$= \quad F(q) - q \cdot f(r) + F^*(f(r)) \tag{86}$$
$$= \quad D_F(q\|r). \tag{87}$$

$\blacksquare$

**Lemma 13** *For any $x$, $r$, $q$ and $\tau > 0$: $\mathcal{S}_\tau(f \circ q, r, x) = -\tau F\left(\frac{r}{\tau}\right) + \tau D_F\left(\frac{r}{\tau}\|q(x)\right)$.*

*Proof:* Immediate from the defintions:

$$
\begin{aligned}
\mathcal{S}_\tau(\boldsymbol{f} \circ \boldsymbol{q}, \boldsymbol{r}, x) &= -\boldsymbol{f}(\boldsymbol{q}(x)) \cdot \boldsymbol{r} + \tau \boldsymbol{f}(\boldsymbol{q}(x)) \cdot \log \boldsymbol{f}(\boldsymbol{q}(x)) & (88) \\
&= -\boldsymbol{f}(\boldsymbol{q}(x)) \cdot \boldsymbol{r} + \tau F^*(\boldsymbol{f}(\boldsymbol{q}(x))) & (89) \\
&= -\tau F\left(\tfrac{\boldsymbol{r}}{\tau}\right) + \tau F\left(\tfrac{\boldsymbol{r}}{\tau}\right) - \boldsymbol{f}(\boldsymbol{q}(x)) \cdot \boldsymbol{r} + \tau F^*(\boldsymbol{f}(\boldsymbol{q}(x))) & (90) \\
&= -\tau F\left(\tfrac{\boldsymbol{r}}{\tau}\right) + \tau D_F\left(\tfrac{\boldsymbol{r}}{\tau} \| \boldsymbol{q}(x)\right). & (91)
\end{aligned}
$$

∎

**Lemma 14** *For any $x$, $\boldsymbol{r}$ and $\tau > 0$:* $\inf_{\boldsymbol{q} \in \mathcal{Q}} \mathcal{S}_\tau(\boldsymbol{f} \circ \boldsymbol{q}, \boldsymbol{r}, x) = -\tau F\left(\tfrac{\boldsymbol{r}}{\tau}\right).$

*Proof:* By Lemma 13 we know that $\mathcal{S}_\tau(\boldsymbol{f} \circ \boldsymbol{q}, \boldsymbol{r}, x) = -\tau F\left(\tfrac{\boldsymbol{r}}{\tau}\right) + \tau D_F\left(\tfrac{\boldsymbol{r}}{\tau} \| \boldsymbol{q}(x)\right)$. Since $D_F$ is nonnegative, yet $D_F\left(\tfrac{\boldsymbol{r}}{\tau} \| \boldsymbol{q}(x)\right) = 0$ when $\boldsymbol{q}(x) = \tfrac{\boldsymbol{r}}{\tau}$, we know the lower bound value $-\tau F\left(\tfrac{\boldsymbol{r}}{\tau}\right)$ is achieved at this point. ∎

## 4 Additional experimental details

### 4.1 Additional Experiment Details: MNIST

In the MNIST experiments we trained a conventional feedforward neural network with a single hidden layer of 512 units and ReLU nonlinearities at the hidden layer. The standard training set of 60K examples was partitioned into the first 55K examples for training and the last 5K for validation. The test set of 10K examples was only used to report the final test results after all hyperparamter tuning was completed on the validation data only. All objectives were trained using the stochastic gradient descent with classical momentum set to 0.9 (i.e. Momentum(0.9)) for 100 epochs.

The hyperparameters and values considered in these experiments were:

learning rate $\in \{0.01, 0.02, 0.05, 0.1, 0.2, 0.5, 1.0, 2.0\}$,
temperature $\tau \in \{0.1, 0.2, 0.5, 1.0, 2.0\}$,
offset $v \in \{0.0, 0.1, 0.2, 0.5\}$,
batch size $\in \{10, 20, 50, 100, 200, 500, 1000\}$, and
combination weights: uniform in the ten value range 0.0 to 1.0 with 0.1 increments.

### 4.2 Additional Experiment Details: CIFAR-10

In all the CIFAR-10 experiments, we trained a Resnet-20 model with layer sizes $(3, 4, 6, 3)$ and filter sizes $(64, 64, 128, 256, 512)$ for 12000 (then 120000; see below) iterations using a TPU with batch size of 128 * 8 = 1024, which corresponds to approximately 49 iterations per epoch for 50000 training examples, or equivalently, 250 (then 2000; see below) epochs total for each run. We used a learning rate of 0.1 with the momentum optimizer with parameter 0.9 along with Nesterov acceleration, along with batch normalization with a decay of 0.9. We also rescaled the squared loss metric by a factor of 0.01 to help stabilize learning. For the expected reward objective, we chose a baseline across $(0, 0.05, 0.1, 0.15, 0.2, 0.4, 0.6, 0.8, 1.0)$. For the composite objective, we found the best surrogate combination using a 0.05 weight on the average of the squared error and reverse imputed kl combined with the 0.95 weight on the expected reward (without any baseline) uniformly across all the bandit feedback tasks.

Although 250 epochs is already substantial training, allowing some objectives to produce good results, to better understand the relative difficulty of the different optimization landscapes we conducted longer training runs of 2000 epochs to ensure convergence was reached by all methods. The results in Table 1 in the main body of the paper were taken from the longer runs to better approximate the training set up used by [3] on the same training data.

### 4.3 Additional Experiment Details: Criteo

There are 35 features used to describe the context and candidates actions on the Criteo counterfactual analysis dataset. Among them, 2 are continuous and the rest are discrete categorical features. We encode the discrete features using one-hot encoding, which results in a 84017-dimensional sparse feature vector for each context $x$. We then build linear models using different loss functions. A

weight vector $W \in \mathbb{R}^{84017}$ is learned for each loss. Different objectives are optimized using SGD with momentum of 0.9. The table below lists the hyper-parameters we tuned for different losses. The final set of hyper-parameters for each method is chosen according to the performance on the validation set.

| Hyperparameters | Values | Methods |
|---|---|---|
| Learning rate | $[0.01, 0.05, 0.1, 0.5, 1.0, 5.0]$ | All |
| Batch size | $[1000, 5000]$ | All |
| $\tau$ | $[0.01, 0.05, 0.1, 0.5, 1.0, 5.0]$ | $\left\|q(\mathbf{x}) - \frac{r-\mathbf{v}}{\tau}\right\|^2, \mathbf{D}_{\mathbf{F}^*}(p\|\pi),$ **Composite** |
| $\lambda$ | $[0.0001, 0.001, 0.01, 0.1, 1.0]$ | POEM |
| $\alpha$ weight of $\mathbf{D}_{\mathbf{F}^*}(p\|\pi)$ | $[0.001, 0.01, 0.1, 1.0]$ | **Composite** |

# 5   Additional experimental results on MNIST

We repeated the experiments on MNIST 10 times to improve significance and to examine learning performance in more detail. Training with the expected reward objective was prone to getting stuck on poor plateaus in the cost sensitive misclassification (full reward feeback) case, so for that objective we repeated the experiments 20 times.

Figure 2a and Figure 2b show the average learning curves (averaged over 10 runs, 20 runs for "expected"), in terms of test misclassification error, for the various objectives. We observe that the expected reward objective is very difficult to optimize, and often gets stuck on a plateau.

(a) Full reward feedback.                    (b) Partial reward (bandit) feedback.

Figure 2: Learning curves (test misclassification error) on MNIST.

To gain a better assessment of the significance of the test misclassification results, Figure 3a and Figure 3b report the test misclassification error averaged over 10 runs (20 runs for "expected") with standard deviations illustrated. These results reinforce the observations made in the main body of the paper, except for the "expected" reward objective, which yielded poor results in the fully observed case. Note that the large error bar for training under expected reward in the cost sensitive classification setting (Figure 3a) is due to training getting stuck on a poor plateau in 6/20 runs. Removing these poor runs and recalculating the mean test misclassification error and standard deviation based on the remaining 14 runs yields the outcome given in Figure 4, which matches the findings in the main body of the paper.

(a) Full reward feedback.

(b) Partial reward (bandit) feedback.

Figure 3: Test misclassification error on MNIST.

Figure 4: Test misclassification error on MNIST, full reward feedback, but for expected reward objective using 14/20 runs that escaped poor plateau.

# 6 Additional experimental results on CIFAR-10

We repeated the experiments on CIFAR-10 10 times to improve significance and to examine learning performance in more detail. Figure 5a and Figure 5b show the average learning curves in terms of test misclassification error, for the various objectives.

(a) Full reward feedback.

(b) Partial reward (bandit) feedback.

Figure 5: Learning curves (training misclassification error) on CIFAR-10.

As above, to gain a better assessment of the significance of the test misclassification results, Figure 6a and Figure 6b report the test misclassification error averaged over 10 runs with standard deviations illustrated. These results reinforce the observations made in the main body of the paper.

(a) Full reward feedback.
(b) Partial reward (bandit) feedback.

Figure 6: Test misclassification error on CIFAR-10.

However, as in the MNIST experiments, we find that after training for 250 epochs direct minimization of the empirical risk $\hat{\mathcal{R}}(\pi)$ is not competitive, yielding both high training and test error in both the fully observed and partially observed reward cases. To investigate whether this training difficulty was caused by plateaus that make it difficult to optimize this objective, we ran the experiments for significantly longer, for 2000 instead of 250 epochs.

In the fully observed case (i.e. cost-sensitive classification), direct empirical risk minimization is eventually able to catch up to the other objectives, achieving both small training and test misclassification error; see Figure 7a. Similarly, for the partially observed case (i.e. contextual bandit), we see a very similar phenonmenon, where direct optimization of empirical risk is able to close the performance gap with the other methods (but does not quite catch up); see Figure 7b. Thus, the hypothesis that the empirical risk objective $\hat{\mathcal{R}}(\pi)$ is indeed difficult to optimize, requiring extended training time and careful tuning to eventually reach competitive results.

We note that in both cases, the results in Figure 7a and Figure 7b significantly improve the results reported for resnet training on CIFAR-10 in [3], using a weaker exploration method in the contextual bandit case here. The main body of the paper also shows an improvement using the same logged data as [3].

(a) Full reward feedback.
(b) Partial reward (bandit) feedback.

Figure 7: Misclassification error on CIFAR-10 data.

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

## COST SENSITIVE CLASSIFICATION

### Batch policy optimization

Assume given **complete** data

|     | $a_1$ | $a_2$ | ... | $a_n$ |
|-----|-------|-------|-----|-------|
| $x_1$ | $r_{11}$ | $r_{12}$ | $r_1...$ | $r_{1n}$ |
| $x_2$ | $r_{21}$ | $r_{22}$ | $r_2...$ | $r_{2n}$ |
| $x_3$ | $r_{31}$ | $r_{32}$ | $r_3...$ | $r_{3n}$ |
| $x_4$ | $r_{41}$ | $r_{42}$ | $r_4...$ | $r_{4n}$ |
| $x_5$ | $r_{51}$ | $r_{52}$ | $r_5...$ | $r_{5n}$ |
| $x_6$ | $r_{61}$ | $r_{62}$ | $r_6...$ | $r_{6n}$ |
| ⋮ | $r_{\cdot1}$ | $r_{\cdot2}$ | $r_{\cdot...}$ | $r_{\cdot n}$ |
| $x_m$ | $r_{m1}$ | $r_{m2}$ | $r_{m...}$ | $r_{mn}$ |

Optimize policy $\pi : X \to \Delta^n$

$$\pi(a\,|\,x) = e^{q(x)_a - F(q(x))}$$
$$F(q(x)) = \log \sum_a e^{q(x)_a}$$
$$q : X \to \Re^n \quad \text{neural network}$$

**Target objective**
- expected reward: $\max \sum_i \mathbf{r}_i \cdot \boldsymbol{\pi}(x_i)$

Done, right?
Not so fast …

**This objective has serious problems**
- actually trying to solve: $\max \sum_i \mathbf{r}_i \cdot \mathbf{f}(q(x_i))$
- plateaus everywhere
- can have **exponentially many** local maxima
- nearly impossible to reach a global optima

**Also: you already know not to train this way!**
to maximize expected reward on **test** contexts

### Recall: supervised classification

Special case: **supervised classification**

|     | $a_1$ | $a_2$ | ... | $a_n$ |
|-----|-------|-------|-----|-------|
| $x_1$ | 0 | 1 | 0 | 0 |
| $x_2$ | 0 | 0 | 0 | 1 |
| $x_3$ | 1 | 0 | 0 | 0 |
| $x_4$ | 0 | 0 | 1 | 0 |
| $x_5$ | 1 | 0 | 0 | 0 |
| $x_6$ | 0 | 1 | 0 | 0 |
| ⋮ | 0 | 0 | 1 | 0 |
| $x_m$ | 0 | 0 | 0 | 1 |

**Target objective**
- expected **accuracy**: $\max \sum_i \mathbf{r}_i \cdot \boldsymbol{\pi}(x_i)$
But you have never trained with this objective
Instead, you used a **surrogate objective**

$$\text{maximum likelihood}$$
$$\max \sum_i \mathbf{r}_i \cdot \log \boldsymbol{\pi}(x_i)$$

**What's going on?**
- $\mathbf{r}_i \cdot \boldsymbol{\pi}(x_i)$ is differentiable, that's not the issue
- training with $\mathbf{r}_i \cdot \log \boldsymbol{\pi}(x_i)$ actually achieves better values of $\mathbf{r}_i \cdot \boldsymbol{\pi}(x_i)$ on the training data

**Useful properties of maximum likelihood**
- $\mathbf{r}_i \cdot \log \boldsymbol{\pi}(x_i)$ is concave in $\mathbf{q}(x_i)$
- it is also calibrated w.r.t. $\mathbf{r}_i \cdot \boldsymbol{\pi}(x_i)$:
$$\forall \epsilon > 0 \exists \delta > 0 \; \mathbf{r} \cdot \log \boldsymbol{\pi}^* - \mathbf{r} \cdot \log < \delta \Rightarrow \mathbf{r} \cdot \boldsymbol{\pi}^* - \mathbf{r} \cdot \boldsymbol{\pi} < \epsilon$$

### Target vs surrogate optimization

Misclassification error on MNIST training data

## COST SENSITIVE CLASSIFICATION

**Definitions**
- Data $\mathcal{D} = \{(x_i, r_i)\}_{i=1}^T$, where $r_i \in \mathbb{R}^K$ specifies reward for each action in context $x_i$
- *True risk* of a policy is $\mathcal{R}(\pi) = -\mathbb{E}[\pi(x) \cdot r]$
- *Empirical risk* on data set $\mathcal{D}$ is $\hat{\mathcal{R}}(\pi, \mathcal{D}) = -\frac{1}{T} \sum_{(x_i, r_i) \in \mathcal{D}} \pi(x_i) \cdot r_i$

**Note** policies normally represented with composition $\pi(x) = f(q(x))$ where $f(q) = e^{q - F(q)}$ with $F(q) = \log(\mathbf{1} \cdot e^q)$

**Theorem** Even for a linear model $q(x) = W\phi(x)$, the function $r \cdot f(q(x))$ can have exponentially many local maxima in $W$

**Motivation** To get around this problem, need to consider *surrogate* training objectives

### Calibrated convex surrogate

**Definitions**
- *Minimal risk* is
$$\mathcal{R}^*(r, x) = \inf_{\pi \in \mathcal{P}} \mathcal{R}(\pi, r, x) = \inf_{q \in \mathcal{Q}} \mathcal{R}(f \circ q, r, x)$$
- *Loss* $L^*(r, x) = \inf_{q \in \mathcal{Q}} L(q, r, x)$ is *calibrated* w.r.t. $\mathcal{R}$ if:
$\exists$ function $\delta(\epsilon, x) \geq 0$ s.t. $\forall \epsilon > 0, x \in X, r \in \mathbb{R}^K, q \in \mathcal{Q}$:
$$L(q, r, x) - L^*(r, x) < \delta(\epsilon, x) \Rightarrow \mathcal{R}(f \circ q, r, x) < \mathcal{R}^*(r, x) + \epsilon$$
- *Smoothed risk* is
$$\mathcal{S}_\tau(\pi, r, x) = -r \cdot \pi(x) + \tau \pi(x) \cdot \log \pi(x)$$
- Let $\tilde{\pi}_\tau = \arg\min_{\pi \in \mathcal{P}} \mathcal{S}_\tau(\pi)$

**Proposition** $\tau < \epsilon / \log K$ implies $\mathcal{R}(\tilde{\pi}_\tau) < \mathcal{R}^* + \epsilon$

**Proposition** Local smoothed risk is equivalent to
$$\mathcal{S}_\tau(\pi, r, x) = -\tau F(\tfrac{r}{\tau}) + \tau D_F(\tfrac{r}{\tau} \| q(x))$$

**Theorem** The surrogate objective
$$L(q, r, x) = \tau D_F \left( q(x) + \tfrac{v}{\tau} \Big\| \tfrac{r}{\tau} \right) + \tfrac{\tau}{4} \left\| q(x) - \tfrac{r - v}{\tau} \right\|^2.$$ is strongly convex in $q$ and calibrated w.r.t. $\mathcal{S}_\tau(f \circ q, r - v, x)$ with $\delta(\epsilon, x) = \epsilon$

### Experimental evaluation

#### Comparing objectives

## BATCH CONTEXTUAL BANDITS

### Coping with missing data

Optimize policy $\pi : X \to \Delta^n$

**Example**
*importance corrected expected reward*
$$\max \sum_i \frac{\pi(a_i \,|\, x_i)}{\beta_i} r_i$$
where $\beta$ are proposal probabilities from behavior strategy

We already know this is a poor objective but what about missing data inference?

**Equivalent** to $\max \hat{r} \cdot \pi$ using
$$\hat{r}_i = \mathbf{1}_{a_i} \tfrac{r_i}{\beta_i}$$

**That is**
- exaggerate observed values by $1/\beta_i$
- fill in all unobserved values with $0$

**This is a pretty lame inference principle
But** … its unbiased!
$$\mathbb{E}[\hat{\mathbf{r}} \,|\, x] = \sum_a \beta_a \mathbf{1}_a \tfrac{r_a}{\beta_a} = \sum_a \mathbf{1}_a r_a = \mathbf{r}$$

### Missing data inference

**Improvement**
"doubly robust estimation"
- instead of filling in with 0s
- fill in with guesses from a model $\mathbf{q}(x)$
$$\hat{r} = \tau q + \lambda \mathbf{1}_a (r - \tau q_a)$$

**Also unbiased**
- as long as $\lambda = 1/\beta_i$
but still alters observed data

**Where should the model come from?**
- could use a separate critic
- train via least squares, then optimize $\pi$
- works okay, but not great

**Note**
- there is only one action value function for single-step decision making, $r(x, a)$
- actor-critic approaches trivialized

### Unified approach

**Unified approach**
- actor and critic are same model
- $\pi = e^{q - F(q)}$ where $F(q) = \log \mathbf{1} \cdot e^q$
- use logits $\tau q(x)$ to predict rewards
$$q(x, a) \approx \frac{r(x, a)}{\tau}$$

**Can combine with previous objectives**
- $KL(\pi \| \hat{\mathbf{p}})$ where $\hat{\mathbf{p}} = e^{\hat{r}/\tau - F(\hat{r}/\tau)}$
- $KL(\hat{\mathbf{p}} \| \pi)$
- $KL(\pi \| \hat{\mathbf{p}}) \leq KL(\hat{\mathbf{p}} \| \pi) + \tfrac{1}{4} \| \hat{r}/\tau - q \|^2$
these are somewhat sensitive to ranking, unlike least squares

**Empirical Bayes estimation**
- optimize hyperparameters $\mathbf{q}$ (neural network)
- integrate out parameters $\xi$

**Example**
marginal likelihood
$$-\log p(r_0 \,|\, a_0, \mathbf{q})$$
$$= -\log \int p(r_0 \,|\, a_0, \xi) p(\xi \,|\, \mathbf{q}) \, d\xi$$
$$= \tfrac{1}{2\sigma^2}(\phi(a_0) \cdot q - r_0)^2 + \tfrac{1}{2}\log \sigma^2 + c$$
- essentially least squares regression

**Can alternatively use surrogates**
$$\min KL(\text{prior} \| \text{posterior})$$
$$\min KL(\text{posterior} \| \text{prior}) \approx \min I(\xi; r_0)$$

## BATCH CONTEXTUAL BANDITS

**Reward estimation** For $x$, $a$, $r_a$, parameters $\lambda(x, a)$, $\tau$, estimate
$$\text{full } \hat{r}(x) = \tau q(x) + \mathbf{1}_a \lambda(x, a)(r_a - \tau q(x)_a)$$

### Surrogate objective

**Definition** Optimal imputed local risk and suboptimality gap
$$\mathcal{S}_\tau^*(\hat{r}, x) = \inf_{q \in \mathcal{Q}} \mathcal{S}_\tau(f \circ q, \hat{r}, x), \; \mathcal{G}_\tau(\pi, \hat{r}, x) = \mathcal{S}_\tau(\pi, \hat{r}, x) - \mathcal{S}_\tau^*(\hat{r}, x)$$

**Proposition** $\forall q, \tau > 0, (x, a, r_a) : \tau D_F\left(\tfrac{\hat{r}(x)}{\tau} \Big\| q(x)\right) = \mathcal{G}_\tau(f \circ q, \hat{r}, x)$

**Theorem** $\forall q, \tau > 0, (x, a, r_a), v$:
$$L(q, \hat{r}, x) \geq \tau D_F\left(\tfrac{\hat{r}(x)}{\tau} \Big\| q(x) + \tfrac{v}{\tau}\right) = \mathcal{G}_\tau(f \circ q, \hat{r}, x) \geq 0$$
$L$ calibrated w.r.t. $\mathcal{S}_\tau(f \circ q, \hat{r} - v, x)$

**Optimization** Given $\mathcal{D} = \{(x_i, a_i, r_i, \beta_i)\}$, context, act, reward, prob
$$\min_{q \in \mathcal{Q}} \hat{L}(q, \mathcal{D}) \quad \text{where} \quad \hat{L}(q, \mathcal{D}) = \tfrac{1}{T} \sum_{(x_i, a_i, r_i, \beta_i) \in \mathcal{D}} L(q, \hat{r}, x_i)$$

### Analysis

**Definition** Expected smoothed risk quantities we seek to control:
$$\mathcal{S}_\tau(\pi) = \mathbb{E}[\mathcal{S}_\tau(\pi, r, x)], \mathcal{S}_\tau^* = \inf_{q \in \mathcal{Q}} \mathcal{S}_\tau(f \circ q), \mathcal{G}_\tau(\pi) = \mathcal{S}_\tau(\pi) - \mathcal{S}_\tau^*$$

**Theorem** For any $q, \hat{r}$ such that $\mathbb{E}[\hat{r}|x] = \mathbb{E}[r|x]$, and baseline $v$:
$$\mathbb{E}[L(q, \hat{r}, x)] \geq \mathbb{E}\left[\tau D_F\left(\tfrac{\hat{r}(x)}{\tau} \Big\| q(x) + \tfrac{v}{\tau}\right)\right] \geq \mathcal{G}_\tau(f \circ q) \geq 0.$$

**Lemma** $\forall \tau, \delta > 0 \exists$ constant $C$ s.t. w.p. at least $1 - \delta$:
$$\mathbb{E}\left[D_F\left(\tfrac{\hat{r}(x)}{\tau} \Big\| q(x)\right)\right] \leq \hat{D}_F(q, \mathcal{D}) + \tfrac{C}{\sqrt{T}} \quad \forall q \in \mathcal{H}.$$

**Theorem** $\forall v, \tau, \delta > 0, \exists C$ s.t. w.p. at least $1 - \delta$:
if $\hat{L}(q, \mathcal{D}) < \tfrac{\tau C}{\sqrt{T}}$ for $q \in \mathcal{H}$ then $\mathcal{G}_\tau(f \circ q) \leq \tfrac{2\tau C}{\sqrt{T}}$

### Experimental evaluation

#### Comparing objectives

MNIST     CIFAR10

**Criteo:** Est. reward on test

| Objectives | $\hat{\mathcal{R}}(\tau) \times 10^4$ |
|---|---|
| Random | $43.68 \pm 2.17$ |
| Behavior | $53.55$ |
| DRO $\hat{\mathcal{R}}(\tau)$ | $53.07 \pm 2.27$ |
| POEM | $51.89 \pm 1.73$ |
| $\hat{\mathcal{R}}(\tau)$ | $51.72 \pm 1.42$ |
| $\|q - \tfrac{r - v}{\tau}\|^2$ | $52.00 \pm 1.28$ |
| $D_F(\hat{p} \| \pi)$ | $52.30 \pm 0.83$ |
| Composite | $55.09 \pm 2.86$ |

Continuous action MNIST

Sum of squared test error on continuous action MNIST ($a \in \Re^{10}$)

[1]Google Brain, [2]University of Alberta

Mailto: schuurmans@google.com