[Reviews · NeurIPS 2019]

Reviewer 1



Summary: The main points in the paper are: -- expected reward objective has exponentially many local maxima -- smooth risk and hence, the new loss L(q, r, x) which are both calibrated can be used and L is strongly convex implying a unique global optimum. -- experiments with cost-sensitive classification on MNIST and CIFAR -- batch contextual bandits: generalized reward imputation model, and showing that a particular surrogate loss which is strongly convex (under some imputation models) tends to perform better empirically. Originality: The work is original. Clarity: The paper is clear to read, except some details in the experimental section, on page 4, where the meanings of the risk R(\pi) is not described clearly. Significance and comments: First, in the new objective for contextual bandits, the authors mention that this objective is not the same as the trust-region or proximal objectives used in RL (line 237), but how does this compare with the maximum entropy RL (for example, Harrnoja et.al, Soft Q-learning and Soft actor-critic) objectives with the same policy and value function/reward models? In these maxent RL formulations, an estimator similar to Eqn 12, Page 5 is optimized. I would appreciate some discussion on this connection. Second, in both these cases, can we provide some guarantees on the test-time performance and possibly some guarantees outside of the seen (x, a, r_a) tuples? The model q is supposed to generalize over these tuples, but can we say something concretely about the performance of the policy learned using the proposed objective? Empirically, the proposed objectives perform a bit better than the other baselines and outperform methods like POEM on standard benchmarks. Overall, I like the idea of looking at surrogates for learning from logged data in a bandit setting, but I find some discussion on how this relates to existing works missing -- for example -- regularized MDPs (entropy, or generally, Bregman divergence regularized MDPs). Also, some discussion on how the proposed objective would "generalize" during training, and what considerations we should keep in mind at test time when using their approach would be appreciated. What I mean by this is that some discussion on the guarantees provided on the expected return of the learned policy under an arbitrary context distribution and action distribution of learned policy would be interesting (E_{x \sim p(x), a \sim \pi(x)} [r(x, a)]) and not just the data distribution ((x, a, r_a) \in D).

Reviewer 2



Originality: I find the idea of using a surrogate objective instead of the expected reward very interesting and novel. the extension to the partially observable setting is interesting as the proposed form finds a common denominator to multiple estimators, but its underlying idea is not novel. Clarity: the paper is overall very well written, and it has a nice flow. Although I am not very familiar with the topic, I certainly enjoyed reading this paper. The only comment is that it might be good to highlight more clearly the form of the convex surrogate. Overall, this seems to me to be a good paper, and my only main concern is the experimental protocol used and the presentation of the results. Specifically: - in Sec 2.2 the decision to show only 100 epochs is arbitrary. I would prefer if you could show the learning curves instead - All results (Fig 1 and 2, and Table 1) should include std or equivalent - Visualizing Train and Test error with the same scale does not really make sense, either use two different scales or separate the plots. - The last sentence of Sec 2.2. is crucial to your finding. I would strongly encourage to replace Fig 1 and 2 with learning curves instead and run them until convergence. Then any difference in optimization landscape will be visible as a change in the integral of the curves. - Similar comments about learning curves apply for Sec 3.6 - It is unclear to me if the reward estimation algorithm is actually evaluated in the experiments. Could you clarify? (it would be nice to include results) - Can you comment on the increased variance demonstrated by Composite on Table 2? Additional comments: - Abstract. "Here we ..." sounds a bit strange since you already list other contributions. - Generally speaking, I find curious that in a paper talking about policy optimization all the experiments consists of classification tasks "reworked" to be decision making. I wonder if it wouldn't be more interesting to use other decision-making benchmarks. - It might also be valuable in Sec 3.6 to run experiments with different data size and distributions other than uniform. - The code was available only for the CIFAR-10 experiments, and not for the MNIST --- Based on the rebuttal, I am increasing my score. The learning curves in the rebuttal might benefit from using log-axis.

Reviewer 3



Detailed comments: The paper considers policy learning for cost-sensitive classification and contextual bandits problems. The focus is on arguing that directly minimizing the empirical risk is difficult if the policy takes the specific form of \pi(x) = f(q(x)) where x is the context, q an unconstrained model and f the softmax function. To reduce the difficulty, surrogate losses are designed which have the “calibrated” property, which means efficient minimization of the surrogate functions implies efficient minimization of the risk. In the cost-sensitive case, the surrogate is convex. Originality: to the best of my knowledge, the two surrogate functions are original. They connect the cost-sensitive classification and contextual bandit which provide insights to both classes of problems. Theorem 1 is a nice result extending the understanding of the difficulty of optimizing empirical risk. Quality: The main contribution of this paper is theoretical and it is of high quality. One technical question is what the role of the baseline v plays in the theoretical result. It is shown empirically beneficial to have such a shift, could the authors provide some theoretical intuition on why this is the case? Empirically, how is v chosen? Clarity: The paper is mostly clear to read. One thing that can be improved is that the F function, first appeared in Proposition 2, is not defined. Later it is defined in the context for KL divergence but it is not clear whether in Proposition 2 it refers to the KL divergence version or not. Significance: I think the contribution is significant to the community which suggests new surrogate objective for both cost-sensitive classification and contextual bandits. Minor comments: Equation (8) has a typo, the middle expression, inside the parenthesis, should be a minus sign instead of a plus sign. Line 127: \hat{R}(\pi) does not yield the highest test error in MNIST. Am I missing something here?

[Author Response · NeurIPS 2019]

**Reviewer #1**   Thanks for the comments!   $\hat{\mathcal{R}}(\boldsymbol{\pi})$ was defined in Equation (1), but we will make the reference clearer.

We should clarify that the theoretical results already consider out-of-sample generalization. Theorem 8 gives a proper form of generalization bound: with high probability, achieving a small value of the empirical surrogate $\hat{L}(\boldsymbol{q}, \mathcal{D})$ guarantees a bound on the **true** suboptimality gap $\mathcal{G}_\tau(\boldsymbol{f} \circ \boldsymbol{q})$. The suboptimality gap $\mathcal{G}_\tau$ in the smoothed risk $\mathcal{S}_\tau$, defined in (14), is defined in terms of full expectations over the underlying distributions $(x, \boldsymbol{r}) \sim p(x, \boldsymbol{r})$ and $a \sim \boldsymbol{\pi}(x)$. In particular, the proof of Theorem 6 explicitly accounts for how the estimator $\hat{\boldsymbol{r}}$ behaves under such a full expectation.

There are connections between this work and entropy regularized RL, but there are also distinctions. To be brief: In our scenario, entropy regularization leads to the smoothed risk (5). The connection between smoothed risk and the KL divergence (8) is known. Reversing the KL, as in (9) (without the quadratic), yields a version of maximum entropy inverse RL. A key novelty in this paper is to augment the maxent inverse RL objective with the quadratic, to achieve an *upper bound* on the original KL (8) that remains *calibrated*. In fact, (9) is the key objective we analyze, which is different from (12). This allows us to prove new generalization bounds in the form of Theorem 8. We are also able to achieve successful empirical results for **batch** policy optimization, which remains a challenge in the sequential case (Fujimoto et al. ICML-19). Finally, most work on entropy regularized RL uses split actor-critic models (or considers policy improvement in full MDP planning), whereas we achieve success with a single model that serves as both.

**Reviewer #3**   Thanks for the comments!   We appreciate the criticism of the experimental exposition, and will improve it as suggested, including adding learning curves and standard deviations. Below are the more detailed experimental results (in terms of test misclassification) for CIFAR10, in both the fully observed (§2.2) and partially observed (§3.6) cases. We also have the same suite of results prepared for MNIST, and standard deviations for Table 1.

(a) CIFAR10 fully observed   (b) CIFAR10 partially observed   (c) CIFAR10 fully observed   (d) CIFAR10 partially observed

*"It is unclear to me if the reward estimation algorithm is actually evaluated in the experiments."* Yes, Section 3.6 used reward estimation (10), with $\lambda$ explained in Line 257 and other hyperparameter choices explained in Appendix 4.

*"Can you comment on the increased variance demonstrated by Composite on Table 2?"* To produce Table 2, hyperparameters were only chosen to maximize estimated validation reward. We do not yet know whether explicitly considering variance on validation will lead to greater statistical separation between methods.

*"I find curious that [...] all the experiments consists of classification tasks "reworked" [...]."* This is inaccurate: the Criteo dataset is a benchmark in this area, which has been extracted from a real online advertising challenge.

*"It might also be valuable [...] to run experiments with different data size and distributions other than uniform."* Yes we agree, but to clarify: Table 1 already gives results on CIFAR10 with different training set sizes and non-uniform data collection [12]. The Criteo results in Table 2 are also based on data gathered by a non-uniform logging policy [16].

*"Code"* Yes, we plan to release the remaining code. *"Compare to off-policy RL & BayesOpt"* The paper is already performing off-policy RL, albeit for single-step tasks given batch data. The batch scenario implies no exploration, only exploitation. A connection to general RL is given above. We will try to squeeze something in about BayesOpt.

**Reviewer #4**   Thanks for the comments!   The baseline $v$ is chosen as a hyperparameter on validation. It has no effect on the global optima under full expectation, but affects the variance of the empirical estimates.

$F$ was actually defined in Line 61, and its use in Proposition 2 was consistent with this definition and subsequent occurrences. We can see how this definition was easily missed however, and we will seek to better highlight it.

Thanks for catching the typo in (8). *Line 127:* You are right, the statement should be "$\hat{\mathcal{R}}(\boldsymbol{\pi})$ yields the highest training error on both MNIST and CIFAR10, the highest test error on CIFAR10, and the second highest test error on MNIST."

As far as we are aware, *ImageNet* has never been used as a testbed in this area. It is quite expensive and would only add another "classification" data set to the evaluation. A non-classification based domain might be more interesting.

[Meta-Review · NeurIPS 2019]

This paper was on the borderline, and generated significant discussion. The meta-reviewer ended up reading the paper in detail, and decided to recommend accept. Please carefully read the reviewers' comments in revising your paper. P.S. The Meta-Reviewer consulted with the authors of POEM regarding previously observed discrepancies in empirical performance (e.g., from Ma et al.), and confirmed that POEM can suffer from instability when learning on datasets with very wide propensity scores.